



# Drivers of the fungal spore bioaerosol budget: observational analysis and global modelling

Ruud H. H. Janssen[1,a], Colette L. Heald[1], Allison L. Steiner[2], Anne E. Perring[3,4], J. Alex Huffman[5], Ellis S. Robinson[4,b], Cynthia H. Twohy[6], Luke D. Ziemba[7]

[1.] Department of Civil and Environmental Engineering, Massachusetts Institute of Technology, Cambridge, MA 02139, USA
[2.] Climate and Space Sciences and Engineering, University of Michigan, Ann Arbor, MI 48109, USA
[3.] Department of Chemistry, Colgate University, Hamilton, NY 13346, USA
[4.] Cooperative Institute for Research in Environmental Sciences, University of Colorado Boulder, Boulder, CO 80305, USA
[5.] Department of Chemistry and Biochemistry, University of Denver, Denver, CO 80208, USA
[6.] NorthWest Research Associates, Redmond, WA 98052, USA
[7.] NASA Langley Research Center Hampton, VA, USA
[a.] now at: TNO Climate, Air and Sustainability, Utrecht, the Netherlands
[b.] now at: John Hopkins University, Dept. of Environmental Health and Engineering

*Correspondence to*: Ruud Janssen (ruud.janssen@tno.nl) and Colette Heald (heald@mit.edu)

**Abstract.**

Bioaerosols are produced by biological processes and directly emitted into the atmosphere, where they contribute to ice nucleation and the formation of precipitation. Previous studies have suggested that fungal spores constitute a substantial portion of the atmospheric bioaerosol budget. However, our understanding of what controls the emission and burden of fungal spores on the global scale is limited. Here, we use a previously unexplored source of fungal spore count data from the American Academy of Allergy, Asthma, and Immunology (AAAAI) to gain insight into the drivers of their emissions. First, we derive emissions from observed concentrations at 66 stations by applying the boundary layer equilibrium assumption. We estimate an annual mean emission of $62\pm31$ m$^{-2}$ s$^{-1}$ across the USA. Based on these pseudo-observed emissions, we derive two models for fungal spore emissions at seasonal scales: a statistical model, which links fungal spore emissions to meteorological variables that show similar seasonal cycles (2 m specific humidity, leaf area index and friction velocity), and a population model, which describes the growth of fungi and the emission of their spores as a biological process that is driven by temperature and biomass density. Both models show better skill at reproducing the seasonal cycle in fungal spore emissions at the AAAI stations than the model previously developed by Heald and Spracklen (2009) (referred to as HS09). We implement all three emissions models in the chemical transport model GEOS-Chem to evaluate global emissions and burden of fungal spore bioaerosol. We estimate annual global emissions of 3.7 and 3.4 Tg yr$^{-1}$ for the statistical model and the population model, respectively, which is about an order of magnitude lower than the HS09 model. The global burden of the statistical and the population model is similarly an order of magnitude lower than that of the HS09 model. A comparison with independent datasets shows that the new models reproduce the seasonal cycle of fluorescent biological aerosol particles (FBAP) concentrations at two locations in Europe somewhat better than the HS09 model, although a quantitative comparison is hindered by the ambiguity in interpreting measurements of fluorescent particles. Observed vertical profiles of FBAP show that





the convective transport of spores over source regions is captured well by GEOS-Chem, irrespective of which emission scheme is used. However, over the North Atlantic, far from significant spore sources, the model does not reproduce the vertical profiles. This points to the need for further exploration of the transport, cloud processing, and wet removal of spores. In addition, more long-term observational datasets are needed to assess whether drivers of seasonal fungal spore emissions are similar across

continents and biomes.

## 1 Introduction

Bioaerosols are omnipresent in the global atmosphere (DeLeon-Rodriguez et al., 2013; Després et al., 2012; Fröhlich-Nowoisky et al., 2016). They contribute to the organic aerosol burden of the atmosphere and therefore can affect weather and climate by influencing cloud and precipitation formation. They can act as ice nucleating particles (INP; Haga et al., 2014; Pratt

et al., 2009; Tobo et al., 2013; Twohy et al., 2016) and can form cloud condensation nuclei (CCN) upon fragmentation in the atmosphere (China et al., 2016; Steiner et al., 2015). Further, bioaerosols can have adverse impacts on human health by acting as pathogens, allergens or toxins (Fröhlich-Nowoisky et al., 2016; Reinmuth-Selzle et al., 2017; Samake et al., 2017) and play a role in the transmission of crop and animal pests (Fisher et al., 2012).

Bioaerosols include bacteria, fungal spore, pollen, and fragments of other organisms, such as plants. The first three groups all

include species that can act as CCN or INP (Fröhlich-Nowoisky et al., 2016), although their activities as cloud nuclei differs per species. The significance of bioaerosol for cloud formation on global and regional scales depends on their abundance, and their relative contribution to INP and CCN populations compared to other aerosol types. On the global scale, their contribution to ice crystal formation is thought to be limited (Hoose et al., 2010; Spracklen and Heald, 2014), although they could still be of importance for cloud formation in specific regions, such as the Amazon (China et al., 2016, 2018; Morris et al., 2014; Pöschl

et al., 2010; Prenni et al., 2009).

Estimates of the emissions of bioaerosols on the global scale vary over almost two orders of magnitude, which prohibits accurate assessment of their impact on cloud formation and air quality. An early estimate that was based on extrapolation of measurements at a few locations was as high as 1000 Tg yr$^{-1}$ (Jaenicke, 2005). Subsequently, global model simulations have been performed that included parameterizations for three main classes of bioaerosols (i.e. pollen, fungal spores and bacteria).

These yielded total emission estimates between 62 and 123 Tg yr$^{-1}$ (Hoose et al., 2010; Myriokefalitakis et al., 2017), with variations between models due to differences in meteorology and land use maps. The variation of estimates for the global bioaerosol burden is large as well, ranging between 121 and 791 Gg resulting from differences between models in emissions, assumed size distributions and formulation of removal mechanisms. However, the emission parameterizations that are incorporated in these models are based on limited observations. All of the above studies used the same emission schemes, or

modified versions thereof. The fungal spore emission scheme (Heald and Spracklen, 2009), referred to as HS09 hereafter, is based on measured concentrations of mannitol, a sugar alcohol that is a proxy for fungal spore concentrations, at a limited number of locations, and simulated emissions of fine and coarse spores from all ecosystems as a function of LAI and specific



humidity. Note that Myriokefalitakis et al. (2017) used a modified form of the HS09 scheme, based on Hummel et al. (2015). An early pollen emission scheme (Jacobson and Streets, 2009) was not based on, or tested against, observations. More recently, pollen emission schemes have been developed based on pollen count observations, and implemented in regional scale models (Wozniak and Steiner, 2017; Zink et al., 2013). Finally, the bacteria emission scheme by Burrows et al. (2009) was developed

by inverse modeling of measured bacteria concentrations over various ecosystems, and assumes constant emissions for each land use type. Since estimates of the global bioaerosol burden strongly depend on their emissions, emission models that are better constrained by observations are urgently needed.

In this work, we focus on fungal spores, as they have a smaller size than pollen, which implies that they are more likely to be transported over longer distances, and to contribute significantly to the organic aerosol budget on the regional and global scale.

They can also produce large quantities of submicron fragments after rupturing in the atmosphere, and thereby contribute to CCN and INP populations (China et al., 2016; O′Sullivan et al., 2015). Fungi emit spores into the air as part of their reproductive strategy. These emissions are thought to depend on temperature and water availability (Boddy et al., 2014; Gange et al., 2007; Jones and Harrison, 2004; Löbs et al., 2020), along with biotic factors. Emissions of spores into the atmosphere can be either active or passive, depending on the species of fungus. Active emission mechanisms include emissions at high

relative humidity with liquid jets or droplets (Elbert et al., 2007; Pringle et al., 2005). Factors that have been proposed to drive the passive emission of fungal spores into the atmosphere include wind (Jones and Harrison, 2004) and rainfall (Huffman et al., 2013; Prenni et al., 2013). Since the sources of fungal spores are diverse, it is challenging to develop a mechanistic description of their atmospheric emissions, and therefore emissions are usually based on extrapolation of the limited number of available observations. These estimated emissions of fungal spores range widely for different methods, including both

models and educated guesses, from 50 Tg year$^{-1}$ (Elbert et al., 2007), 28 Tg year$^{-1}$ (HS09), 186 Tg year$^{-1}$ (Jacobson and Streets, 2009) to 79 Tg year$^{-1}$ (Sesartic and Dallafior, 2011). Moreover, the seasonal cycle in these estimates is either absent, or assumed to be instantaneously related to the seasonal cycle of the driving variables.

In this study, we develop two new schemes for the emission of fungal spores on seasonal time scales, using a previously unexplored source of observed fungal spore concentrations over the United States, and building on available knowledge about

the drivers of their emissions. Subsequently, we implement these emission schemes in the GEOS-Chem chemical transport model to calculate the global emissions and burden of fungal spores. Finally, we evaluate the ability of both emission schemes to simulate spatial and seasonal variations in observed fungal spore concentrations and compare results from the new schemes to those from the previously developed Heald and Spracklen (2009) scheme.

## 2. Developing new emission schemes for fungal spores

In this section, we first describe how we infer fungal spore emissions from observed concentrations, and subsequently we explain how we develop two new emission parametrizations from these derived emissions. The first parameterization is a purely statistical one, and is derived by relating spore emissions to meteorological and land use variables, using multivariate





linear regression. The second parameterization is based on the fact that fungal spore production is the result of a biological process. We aim to represent the production of spores with a simple population model that accounts for the growth of fungi and fungal spores during the year. The overall goal is to obtain emission models that are better constrained and validated by observational data than the existing HS09 model, but that are still simple and straightforward to implement in 3D models.

## 2.1 Fungal spore observations

Our emission scheme is based on multi-annual time series (6 years, from 2003 to 2008) of spore counts at 66 stations across the continental US operated by the American Academy of Allergy, Asthma, and Immunology (AAAAI). Members of the National Allergy Bureau monitor spore and pollen counts at these stations, where samples are collected at least 3 days a week using a Burkard spore trap (Hirst, 1952; Levetin, 2004). Spore traps are situated on an unobstructed rooftop at least one story above ground, with no local spore or pollen sources (http://pollen.aaaai.org/nab). In the Burkard spore trap, air is drawn into a 14 mm x 2 mm orifice at 10 L min$^{-1}$, and any airborne particles with sufficient inertia are impacted on either a greased tape or a greased microscope slide beneath the orifice. The slides are then examined by microscopy for counting and identification of spores. The standard orifice on the Burkard sampler is efficient for particles down to 3.7 µm (Levetin, 2004), which means that the collection efficiency of the smallest spores is less than unity. The reported spore counts therefore represent lower limit values: for the size distribution parameters as defined in Section 3.2, ~40% of the mass concentration and ~83% of the number concentration would fall in the size range for which the collection efficiency is below unity. Without a better understanding of how the collection efficiency varies with size, we cannot assess what fraction of these particles go undetected by the Burkard spore trap.

Specified spore counts are available at the genus level, but for our analysis we only use the total daily spore counts. The observed concentration ranges between 0 and 6.3x10$^4$ m$^{-3}$ for all stations and years with a mean of 5.4x10$^3$ m$^{-3}$. Figure 1 shows a map with an overview of the AAAAI stations used in this analysis (with the exception of Anchorage, AK), and the mean spore concentration over the full length of the measurement period for each station. For 36% of the stations, no observations were available during winter, which has consequences for the derived fluxes during that time of year (see Section 3.1). The map also shows the land use, which is a simplified version of the Olson Terrestrial Ecoregions data set (Olson et al., 2001) and uses the same lumping into broad land use categories as Burrows et al. (2009). The concentrations show no clear relation to land use types, although the 3 stations with the lowest concentrations are located in regions that are dominated by deserts and shrubs.

## 2.2 From concentrations to fluxes

To develop an emission scheme from these observations, emission fluxes need to be derived from measured concentrations first. This derivation consists of two steps: 1) the conversion from concentrations to net surface fluxes and 2) the conversion



from net surface fluxes to emissions fluxes, by subtracting the deposition flux. We describe this procedure here, using Figure 2 to visually present an example at one site.

Rainfall poses a challenge for deriving bioaerosol fluxes. A number of studies (e.g. Geagea et al., 2000; Huffman et al., 2013; Prenni et al., 2013) have demonstrated that rain can act as a trigger for the release of bioaerosols from vegetation and soils.

However, at the same time, wet deposition removes aerosols from the atmosphere. This offsetting effect complicates the relationship between rainfall and net fungal spore fluxes. Therefore, to simplify our analysis, we remove spore counts from our observational dataset that were made on days on which any rainfall occurred (on average 32% of the days at each station), as established by the categorical rain (*crain*) variable in the National Centers for Environmental Prediction (NCEP) North American Regional Reanalysis (NARR) product (Mesinger et al., 2006) dataset. This necessarily prohibits an assessment of

the influence of rainfall on fungal spore emissions on the same day. We note that this is a coarse filtering and that emissions of fungal spores may respond to rainfall on timescales up to three days (Sarda-Estève et al., 2019). Given our focus on 20-day average emissions (see below), we do not apply a more sophisticated treatment, but note that further effort to characterize the relationship between fungal spore emissions and rainfall could inform higher temporal resolution modeling.

There are several methods available for translating atmospheric concentrations to surface fluxes. Here, we apply the

equilibrium boundary layer assumption (Betts, 2000), which states that over sufficiently long periods (at least several days), boundary layer depth over land reflects a statistical equilibrium between surface heating that acts to deepen the boundary layer and subsidence of warm air that acts to decrease boundary layer height. The surface flux can then be calculated from boundary layer concentrations by applying the tracer conservation equation, which accounts for the effects of horizontal and vertical transport. We assume that convection maintains a well-mixed boundary layer, in which scalars, reactants, and aerosols have a

constant profile over the depth of the boundary layer. This method has been used before to infer seasonal $CO_2$ surface fluxes from measured concentrations (Bakwin et al., 2004; Helliker et al., 2004).

The tracer conservation equation in a simplified form, which does not account for horizontal advection, is as follows:

$$F_s = (\langle C \rangle - C_{FT})w_m + h\frac{\partial \langle C \rangle}{\partial t} - C_{FT}\frac{\partial h}{\partial t} \qquad (1)$$

in which $F_s$ is the surface flux ($m^{-2}$ $s^{-1}$), $\langle C \rangle$ the boundary layer concentration of species C ($m^{-3}$), $C_{FT}$ the free tropospheric concentration of C ($m^{-3}$), $w_m$ the subsidence velocity at boundary layer top ($m$ $s^{-1}$), h the well-mixed boundary layer height (m), and t is time. The three terms on the right hand side of Eq. 1 represent the vertical advection, storage, and entrainment terms, respectively.

In our analysis, C is the concentration of fungal spores in the boundary layer as reported at the AAAAI stations. The measurement heights for the AAAAI stations are not specified, but the measurement locations are at least one story above the ground. This means that the sampling locations are in the atmospheric surface layer, which likely leads to an overestimation of the boundary layer concentrations. For instance, Perring et al. (2015) found that PBAP concentrations aloft (up to the 900



hPa level) are only between 5-55% of those at the surface. The concentration of fungal spores in the free troposphere ($C_{FT}$) is not well characterized. Based on the vertical profile of fluorescent bioaerosol concentrations observed in and above the boundary layer over the US western plains (Twohy et al., 2016), we assume that the concentration of spores decreases by about an order of magnitude between BL and FT. Hence, we set $C_{FT}=0.1<C>$. This is clearly a crude assumption and we discuss the

sensitivity of the calculated fluxes to different values of this dilution factor in Section 5.

We take the subsidence velocity from the NARR data, as vertical velocity interpolated to the mean height of the afternoon (12:00-18:00 LT) boundary layer top (Figure 2b). With a spatial resolution of 32 km (about $0.3^0$) and 8 output fields per day (representing 3-hourly averages), NARR output provides a reasonable spatial and temporal match for each of the AAAAI stations of interest. In the boundary layer equilibrium assumption, we take the mean height of the afternoon boundary layer as

the daily boundary layer height (Figure 2c). We assume that the height of the daytime mixed-layer during daytime is representative of the mean boundary layer height for each day, and that the summed depth of the nocturnal boundary layer and the residual layer during night-time is similar to the daytime boundary layer height (Bakwin et al., 2004; Helliker et al., 2004). Williams et al. (2011) found that for $CO_2$, horizontal advection can be of the same order of magnitude as vertical advection. However, for fungal spores, concentration differences in the horizontal are likely smaller compared to those in the vertical,

given their short atmospheric lifetime. Therefore, the impact of vertical mixing may be relatively stronger and the impact of horizontal transport relatively weaker than for $CO_2$, which is better mixed throughout the atmosphere. In Equation 1, horizontal advection is neglected, because there is no reliable way to constrain the horizontal transport of fungal spores.

We use Eq. 1 to calculate running average fluxes over 20 days in order to minimize the effects of synoptic scale variability on the relationship between concentration and flux while maintaining the seasonal cycle (Bakwin et al., 2004). A consequence of

this choice is that the contribution of short-term storage and entrainment effects to the calculated surface flux is minimal (Williams et al., 2011). Figure 2d shows the calculation of the three terms from Equation 1. The vertical advection term contributes most to the calculated net surface flux, and therefore we explore how assumptions related to this term impact derived fluxes in Section 3.4. In contrast, the combined storage+entrainment term becomes negligible in magnitude (<10%) compared to the surface flux for most stations when an averaging period of 20 days is applied (Figure S1) which shows that at

seasonal time scales storage and entrainment contributions can be neglected without introducing large errors in the surface flux calculation. Whether inclusion of horizontal advection in the boundary layer budget equation would substantially impact these results remains an open question. It likely varies per site, depending on whether there are spore sources upwind of the site or not.

As a final step in the derivation of the emission flux of fungal spores, we calculate the dry deposition flux with an offline

version of the aerosol dry deposition scheme that is also used in the GEOS-Chem model (Zhang et al., 2001). To run this bulk deposition scheme, we use meteorological fields from the NARR as input and we assume a mean fungal spore diameter of 2.5 µm (see Section 3.2) and a density of 1 g cm$^{-3}$ (Heald and Spracklen, 2009). The calculated deposition velocities are low (<0.1 cm s$^{-1}$) at all stations and seasons, so the deposition flux is of minor influence in the derivation of the emission flux from the net surface flux (Figure 2e).



The conversion of the fungal spore counts to emission fluxes yields a mean emission of $62\pm31$ m$^{-2}$ s$^{-1}$ over all years and stations, with a strong seasonal cycle. The mean ratio between concentrations and fluxes does not vary substantially between sites and land use types (Figure S2). About a third of the stations (26) are associated with the 'forests' land use type, while other land use types are not as well represented in the dataset (Figure 1). Therefore, for the purpose of developing the emission scheme, we do not distinguish between land use types. Very few flux measurements of bioaerosols in general and fungal spores in particular are available to compare the magnitude of emission that we estimate here. Carotenuto et al. (2017) measured microbial fluxes over a Mediterranean grassland, reporting mean fluxes of $8.3\pm11.1$ m$^{-2}$ s$^{-1}$ in 2008-2010 and $10.6\pm6.2$ m$^{-2}$ s$^{-1}$ in 2015. However, comparison with our derived fluxes is complicated by the fact that they report net fluxes of viable bioaerosols, which represent only a fraction of the total bioaerosol population and are likely composed of both fungal spores and bacteria. Crawford et al. (2014) derived fluorescent bioaerosol fluxes over a Colorado pine forest by applying flux-gradient relationships. Fluorescent clusters that were tentatively associated with fungal spores showed estimated night-time emissions up to 6000 m$^{-2}$ s$^{-1}$ under humid conditions, although they observed net deposition fluxes during much of the rest of the day and under dry conditions. Finally, Ahlm et al. (2010) reported upward fluxes of accumulation mode particles in a tropical forest of up to 5000 m$^{-2}$ s$^{-1}$. They claim that these emitted particles could be fungal spores, although their observations are complicated by dry deposition of particles of supposedly anthropogenic origin. More definitive measurements of spore fluxes would be useful for further comparison with our derived fluxes.

## 2.3 Statistical model for spore emissions

For our initial model, we take a purely statistical approach in quantifying fungal spore emissions at seasonal time scales and perform a multivariate linear regression (MLR) on the derived fungal spore fluxes. For this purpose, we combine the AAAAI data with MERRA2 meteorological data (Gelaro et al., 2017) at $0.5°\times0.625°$ resolution. With our objective of implementing this emission scheme into the GEOS-Chem model, we use MERRA2 meteorology here (as used in GEOS-Chem), rather than the NARR dataset used in Section 2.2. In addition, the NARR archive does not contain some surface variables that are relevant for describing land surface-atmosphere exchange, such as friction velocity and roughness length. For the most important variables in our analysis (temperature at 2 m $T_{2m}$ and specific moisture at 2 m $q_{2m}$), we verify that the MERRA2 and NARR datasets are consistent. We find very good agreement between the two datasets despite different origins and spatial resolutions, with r$^2$=0.94 and NMB=0.0 for $T_{2m}$ and r$^2$=0.92 and NMB=0.03 for $q_{2m}$. For wind speed at 10 m ($U_{10m}$), we do not find good agreement (r$^2$=0.01 and NMB=-0.59), but this variable is less important in our analysis than $T_{2m}$ and $q_{2m}$. Therefore, we conclude that the choice of meteorological dataset does not have a major impact on our analysis.

In addition to MERRA2 data, we use 4-day LAI observations from MODIS (Myneni et al., 2015) aggregated to $0.25°\times0.25°$ resolution as a variable in our regression analysis. The LAI data used here shows good agreement with the LAI used in the GEOS-Chem simulations, with r$^2$=0.80 and NMB=-0.02. We also include time (measured in days from the start of the AAAAI time series) to account for any linear trend in fungal spore emissions, as in Porter et al. (2015). Variables showing a strongly



skewed distribution (e.g LAI and 2m temperature) were log-transformed to fulfill the MLR requirement of normally distributed variables.

In the MLR, the first independent variable is selected based on the $r^2$ score. Subsequently, all other variables are tested and the one that leads to the largest decrease in the Bayesian Information Criterion (BIC) is kept as second independent variable.

This procedure is repeated until all meteorological and land surface variables are evaluated. Finally, we only keep the variables that lead to a significant decrease in BIC for inclusion in the statistical model. The BIC provides a measure of relative model performance, and can be used to find an optimum number of explanatory variables in statistical models, by including a penalty for overfitting (Porter et al., 2015). Unlike the $r^2$, it will not increase whenever a new variable is added, but rather yields a minimum value at which a maximum model skill is reached without including redundant variables.

The regression analysis identifies specific humidity at 2 m ($q_{2m}$), leaf area index (LAI), and friction velocity ($u^*$) as the top independent variables that explain the seasonal cycle in fungal spore emissions (Fig. 3). Figure 3 shows that a minimum in ΔBIC is not reached until after the inclusion of about 6 variables. Given that including this many variables is somewhat impractical and the gain in model skill (represented by $r^2$) by adding additional variables is small, we choose to limit the number of predictors to 3. Several independent variables have similar correlations with the spore emissions, therefore we have

tested the robustness of our variable selection method by forcing different variables as the first variable in the MLR analysis (*LAI* and 2 m temperature $T_{2m}$). In each of these cases, the top 3 of independent variables are a combination of $q_{2m}$, *LAI*, $u^*$ and $T_{2m}$, which gives confidence in the selection of $q_{2m}$, *LAI* and $u^*$ as driving variables in our statistical model. Our statistical emission function is thus:

$$F_{stat} = b_0 + b_1 \cdot q_{2m} + b_2 \cdot LAI + b_3 \cdot u^* \qquad (2)$$

with coefficients $b_0$-$b_3$ as in Table 1 (determined from fitting procedure described in Section 2.5).

This selection does not mean that the chosen variables specific humidity, LAI and friction velocity are in fact the actual drivers of fungal spore emissions on seasonal time scales. Rather, they are variables which show a similar seasonal cycle as, and therefore a statistical relationship with, the emissions over all stations and years, and which can be tentatively associated with

the growth of fungi and the emission of spores. In other words, it seems likely that humidity and vegetation biomass in some form play a role in the growth of fungi and wind speed in the emission of their spores, and it is therefore plausible that the correlations are indicative of the actual underlying mechanisms. Note that the first two variables are the same as identified in the previous fungal spore scheme developed by HS09. Furthermore, we note that other meteorological drivers, including rain which is specifically excluded here, may become important for controlling fungal spore emissions at shorter time scales.

## 2.4 Population model for spore emissions

A model that explains and quantifies the emissions of fungal spores at the seasonal time scale should contain the driving variables of spore emissions at the appropriate time scale. These drivers may include both environmental and biological factors.



In the literature on fungal growth, temperature and moisture are often mentioned as environmental factors that determine fungal fruiting patterns (Boddy et al., 2014; Damialis et al., 2015; Gange et al., 2007; Kauserud et al., 2008), while resource availability and competition are also thought to play a role.

Here, we take a first order approach and assume that fungal fruiting (and subsequent spore production) is a biological process

5 that is temperature driven. Further, we assume that greater vegetation biomass can sustain larger fungal populations, by providing more resources for fungi to thrive on. Hence, we represent fungal growth by a logistic growth model, in which the growth rate is a function of temperature and the carrying capacity a function of LAI:

$$\frac{dN}{dt} = rN\frac{K - N}{K} - mN \tag{3}$$

in which $N$ is the population size (m$^{-2}$), r the growth rate (s$^{-1}$), $K$ the carrying capacity (m$^{-2}$) and $m$ the mortality rate (s$^{-1}$). The mortality term is added to ensure that the fungal population decays when conditions are not suited for growth. The growth rate is represented as follows:

$$r = r_{max}\left(\frac{T_{max} - T}{T_{max} - T_{opt}}\right)\left(\frac{T - T_{min}}{T_{opt} - T_{min}}\right)^{\left(\frac{T_{opt} - T_{min}}{T_{max} - T_{opt}}\right)} \tag{4}$$

in which $r_{max}$ is the maximum growth rate (s$^{-1}$), $T_{max}$, $T_{min}$ and $T_{opt}$ are the maximum, minimum and optimal temperatures for fungal growth (ºC), respectively, and $T$ is the actual temperature (ºC).

The carrying capacity $K$ is assumed to be a linear function of LAI:

$$K = l_1 + l_2 LAI \tag{5}$$

, in which $l_1$ and $l_2$ (m$^{-2}$) are two fitting parameters that determine the sensitivity of $K$ to LAI.

Emissions of spores from the fungi are then modeled as a function of friction velocity, following a saturation function (Carotenuto et al., 2017; Zink et al., 2013):

$$f_{u_*} = \frac{1}{1 + e^{-s_1(u_* - s_2)}} \tag{6}$$

in which $f_{u*}$ is a dimensionless emission factor which is a function of friction velocity u$^*$ (m s$^{-1}$), and in which $s_1$ and $s_2$ are two fitting parameters that determine the sensitivity of $f_{u*}$ to u$^*$.

Finally, the emission flux of fungal spores $F_{pop}$ (m$^{-2}$ s$^{-1}$) is calculated as:

$$F_{pop} = f_{u_*}N \tag{7}$$





An important simplification in this model is the fact that we do not make any distinction between the population size of the fungi and the number of spores that they produce. In principle, this distinction could easily be included in this formulation by separating the number of fungi and fungal spores into two variables in Eq. 3. However, we have no observational constraints on the size and number of fungi, and therefore such a distinction would only increase the number of variables and free parameters in the set of equations, without providing any verifiable results for the fungal population size. An implicit assumption in this model, which is a consequence of not explicitly including a reservoir of spores, is that emissions have no effect on the fungal spore population size.

### 2.5 Model fitting

We fit the statistical model, the population model and the HS09 model to the calculated emission time series for each individual station (Figure 4), using a non-linear least-squares minimization algorithm (Newville et al., 2014). Meteorological fields from MERRA2 were used in this fitting procedure to ensure consistency with the meteorological data that is used to drive atmospheric transport in GEOS-Chem. When we fit the statistical model with $q_{2m}$, $LAI$ and $u^*$ as independent variables to the emission time series, we find that it has reasonable skill in explaining the seasonality of the observation-based emissions, with $r^2$=0.74 and NMB=-0.004 (Figure 4a). Table 1 shows the parameter values for the best fit. The fitted population model captures the seasonal cycle in fungal spore emissions better than the statistical model with $r^2$=0.85 and NMB=0.004. Table 2 shows the fitted parameters for the population model. In essence, spore emissions in the population model follow a delayed response to temperature and LAI, due to the growth and mortality of the fungi in the population model. The friction velocity has only a minor influence on the emissions. Of the three models, the HS09 model, which shares two variables with the statistical model, but has only one regression coefficient (i.e. it is of the form $F_{sp} = c \cdot q_{2m} \cdot LAI$) shows the least skill in representing the timing and magnitude of the seasonal cycle ($r^2$=0.72 and NMB=-0.193). The fitted coefficient $c$ here has a value of $2.9 \times 10^{-8}$ gC m$^2$ s$^{-1}$ ($4.4 \times 10^3$ m$^{-2}$ s$^{-1}$), which is substantially lower than the original value of $5.2 \times 10^{-8}$ gC m$^{-2}$ s$^{-1}$ for the fine mode in HS09. We note that the original HS09 scheme was derived using a much more limited set of mannitol observations. Both the statistical and the HS09 model predict a seasonal cycle which is out of phase with the derived emissions by roughly 1 to 2 months (Figure 4). Some years show two peaks in spore emissions (for instance, there are peaks in June and August-September 2005, and in June and September-October 2008), which are not reproduced by any of the models.

## 3. Integrating fungal spore emissions in a global model

### 3.1 Chemical transport model

We implement our newly developed fungal spore emission schemes in the GEOS-Chem chemical transport model (v11-01; www.geos-chem.org). Simulations are run for two years (2015 and 2016), of which the first year is used for spin up, with an emission and transport time step of 30 and 10 min., respectively. The model is driven by assimilated meteorology from the



NASA Global Modeling and Assimilation Office (GMAO), here using the MERRA2 product (Gelaro et al., 2017). Global simulations are performed at a horizontal resolution of 2 x 2.5 degree and 47 vertical levels. Spore emissions are implemented as a Harvard–NASA Emission Component (HEMCO; Keller et al., 2014) extension, which uses the model meteorology at either the surface or the lowest vertical level, and MODIS LAI product from Yuan et al. (2011) for the year 2008 to calculate

emissions (note that the MODIS product used here is not available for 2016, but we find only a minor difference in LAI between 2008 and 2016 in an offline comparison, and therefore do not expect this to noticeably impact results shown here). The dry deposition and sedimentation of aerosol particles is described by the Zhang et al. (2001) bulk aerosol deposition scheme. We made minor adaptations to this scheme to accommodate sedimentation of bioaerosols as a new coarse aerosol class, in addition to dust and sea salt (see Section 3.2 for discussion of assumed particle size). Wet deposition is treated by the

Liu et al. (2001) scheme, assuming that spores are in the coarse mode. In this scheme, we assume efficient scavenging of fungal spores by rainout and conversion of cloud condensate to precipitation. We address the validity of this assumption in a sensitivity analysis (see Section 5).

In our initial simulations, we found unrealistically high fungal spore concentrations in winter for several locations in the US and Europe in our new schemes (see Section 3.5). This is the result of the interplay between low but steady emissions in winter

and a (lack of) wet deposition for the simulated year. Since the AAAAI data show gaps for many stations in winter and our observational analysis does not explicitly take into account wet removal, it is likely that our emission schemes are not representative of winter conditions. Therefore, we apply a 2 m temperature threshold of 0°C, below which there is no emission of fungal spores. This value corresponds to the minimum temperature for fungal growth as derived for the population model, and it makes sense physically to not have emissions from frozen surfaces. Since the emissions in winter are already low, this

threshold does not affect the global budget substantially, while improving the simulated seasonal cycle significantly (Section 3.5). Note that this threshold is only applied to our new schemes and not to the original HS09 scheme to which we compare.

### 3.2 Size distribution

The assumed geometric mean diameter ($D_p$) and standard deviation ($\sigma$) of the size distribution of fungal spore is central in linking their number concentration to mass concentration and for calculating dry and wet deposition. Previous studies made

different assumptions on the size distribution of fungal spores. HS09 assumed two modes: a fine ($0<D_p<2.5$ μm) and a coarse ($2.5<D_p<10$ μm) mode with a geometric standard deviation $\sigma$ of 1.59 (Spracklen and Heald, 2014), while Hoose et al. (2010) and Myriokefalitakis et al. (2017) applied a monodisperse distribution with diameters of 5 μm and 3 μm, respectively. Here, we constrain the fungal spore size distribution by using WIBS observations in regions of the US that are thought to be dominated by fungal spores from a recent campaign (Fig. 5). The campaign was conducted in summer of 2016 on a NOAA

Twin Otter aircraft using a WIBS-4A from Droplet Measurements Technologies. Operations were based out of Mobile, AL (June 11–16), Asheville, NC (June 16-23), and Madison, WI (June 23-29) to target latitudinal differences in fluorescent particle sources and distributions. The inlet and flight conditions were selected specifically to allow sampling of coarse-mode aerosol (>80% transmission for sizes below 5.4 μm dropping to 35% at 10 μm) and data was analyzed using the seven-type





methodology presented in Perring et al. (2015). To extract "fungal-like" concentrations and size distributions, we include type A, AB and ABC fluorescent particles with optical sizes between 1 and 5 µm. The size distributions from the 2016 campaign were nearly identical to those reported in Perring et al. (2015) for the same fluorescent particle types in the Eastern US. The parameters for the ambient distributions are similar across a wide band of latitudes, so we have chosen to use a $D_p$ of 2.5 µm

and a $\sigma$ of 1.5. These ambient size distribution parameters are generally in good agreement with size distributions for known fungal spore cultures in the laboratory, although the lab distributions for individual species are somewhat narrower with $1.2 < \sigma < 1.4$, which may be related to spores being mixed and aged in the atmosphere. As in previous studies (Heald and Spracklen, 2009; Sesartic and Dallafior, 2011), we assume a fungal spore density of 1 g cm$^{-3}$.

## 3.3 Global emissions and burden

We implement both the population model and the statistical model in GEOS-Chem to calculate global emissions and burden of fungal spores and compare these results to those of the HS09 scheme. Both the statistical model and the population model produce emissions that are about an order of magnitude lower (3.7 and 3.4 Tg yr$^{-1}$, respectively) on the global scale than the HS09 scheme (31 Tg yr$^{-1}$; note that we implement the scheme with the original coefficients in GEOS-Chem, and not the optimized version as in Section 2.4). The HS09 scheme total spore emission of 31 Tg year$^{-1}$ (of which 8 Tg yr$^{-1}$ are in the fine

mode and 23 Tg yr$^{-1}$ in the coarse mode, following sizes specified in that study) is 10% higher in the current implementation than in the original study. This difference is due to different model meteorology (GEOS-4 versus MERRA2), LAI and year of simulation. Despite the slightly higher emissions in our simulations, we find that the burden is about 30% lower than in the original study, due to more efficient wet deposition of coarse particles in the newer model version. Similar to the emissions, the burdens for the statistical and population model are also about an order of magnitude lower than the burden for the HS09

scheme. The fungal spore lifetime for the statistical model is lower than for the population model (1.4 vs. 2.1 days), because the statistical model emissions are more concentrated in regions that are characterized by high rainfall (i.e. the tropics), and therefore with faster wet removal of particles. An overview of global spore emissions, burden and lifetime for the three schemes as implemented in GEOS-Chem is given in Table 3.

All three emission schemes yield a similar spatial pattern of annual mean emissions with emission peaks across the tropics and

minor peaks in the southeastern US, Europe and south-east Asia (Figure 6). This similarity is not surprising, as all schemes use LAI as input, and in the tropics high temperatures accompany high specific humidity. The seasonal cycles in emissions and burdens, however, show more pronounced differences between the schemes (Figure 7). Over North America and Asia, for instance, emissions from the statistical and the HS09 model peak in July while those of the population model peak in August. These differences in emissions are reflected in the concentrations. Over North America, peak concentrations of spores from

the statistical and the HS09 model peak one month after the emissions in August, but spores from the population model concentrations peak in September, with a secondary peak in November. These delays between emissions and concentrations are mainly caused by the occurrence of wet deposition; in months when high emissions coincide with high rainfall, the resulting concentrations may be lower than in months with somewhat lower emissions, but also with lower amounts of precipitation.





Also in Europe, the population model emissions start increasing later than in the other two schemes (May versus April), but when they increase it happens more rapidly. Over Asia, simulated concentrations from the statistical and the HS09 model follow quite different seasonal cycles than the population model, with the former two peaking in August and the latter in November. This is a consequence of the interplay between emissions and wet deposition: rainfall maxima occur in July and

August in this region, related to the East Asian monsoon. Statistical model emissions show a strong peak during the same period, and therefore statistical model concentrations are still high. Population model emissions, on the other hand, are much weaker and therefore wet deposition has a strong influence on its concentration in summer.

Over South America, the statistical model predicts a stronger seasonal cycle in emissions than the population and the HS09 model, and also the timing differs, with the emissions from the statistical model showing a minimum in July and the other

models in June. As a results of these different seasonal cycles in emissions, all models show different seasonal cycles in the concentrations. The statistical model shows peak concentrations from April through August, the population model peaks in July and August and the HS09 model in April. The statistical and population model yield minima during the transition period from the dry to the wet season and the wet season (October-February), while the HS09 model shows minimum concentrations in June. Due to the strong emissions of the HS09 model, emissions and concentrations have similar cycles. Since wet deposition

in GEOS-Chem is size-dependent, it has a stronger influence on the spore concentrations from the HS09 scheme, due to the presence of a fine and a coarse mode (see Section 3.2). For the other two emission models, the modeled concentrations clearly result from the interplay between emissions and wet deposition during the seasons.

As a verification of our implementation of these emission schemes, we compare the results of both schemes within the GEOS-Chem simulation to the AAAAI data from which they were developed, as well as the HS09 scheme. Theoretically, one would

expect near-perfect agreement here, but there are several factors, largely related to comparing a single observation with gridbox average values, which can degrade this comparison. First, in GEOS-Chem, each 2x2.5° grid box can contain multiple land cover types, including land use types, like water surfaces, from which no spores are emitted. Including these land cover types would lead to an underestimate in grid box average spore emissions compared to emission at the AAAAI station in that grid box, which has been shown before to be an issue in the model-measurement comparison of deposition (Silva and Heald, 2018).

To be able to make a fair comparison between grid boxes and point measurements, we run a simulation in which the grid boxes had a fully emitting land cover. Further, for the comparison of modeled and observed spore concentrations, several additional factors contribute to model vs. point measurement differences, including the exclusion of days with rain and wet deposition in the offline calculations, and in general differences in meteorology between the years of observations and simulation (2003-2008 vs. 2016).

We find that GEOS-Chem is able to reproduce the broad pattern in annual average fungal spore emissions over the US, with high emissions in the east and low emissions in the west, for both the emissions from the statistical and the population model. The HS09 scheme also reproduces this pattern, but with a strong overestimation of emissions over the whole US (NMB=10.1), as expected given the order of magnitude difference in emissions. We note that this overestimate is largely driven by the inclusion of the coarse mode emissions in HS09, which make up 75% of the emissions based on the mannitol observations





used to constrain that model. However, the observed size distribution data (see Section 3.2) seems inconsistent with this preponderance of fungal spores in the coarse mode; more work is needed to understand the size distribution of fungal spores and the efficiency with which spores are sampled by various measurement techniques. For the statistical and the population model, we find that the GEOS-Chem emissions have a small negative bias (NMB=-0.01 and -0.08, respectively), but that the

skill in reproducing seasonal variations at the AAAAI stations is low ($r^2$=0.28 and 0.26, respectively) (Figure 8). The latter can be explained by the fact that, although the combined seasonal cycle over all stations is reproduced well in the model fit (Fig. 4), the emission models do not capture the variations between stations that may result from, for instance, different land use (and vegetation) types surrounding the stations.

We can conclude that the statistical model reproduces the magnitude and seasonal cycle of fungal spore emissions slightly

better than the population model. We explore comparisons against independent measurements in Section 3.4 to identify whether one scheme has additional model skill over the other.

## 3.4 Validation with independent datasets: seasonal cycle and vertical profile

Since our models are based on observed spore counts from the United States, a validation with independent data sets is vital,

particularly for other regions. Unfortunately, there are limited direct observations of fungal spore concentrations. Measurements of fluorescent biological aerosol particles (FBAP) are available that can, in principle, be used for this purpose. However, caution is needed in the comparison with spore concentrations, because the fluorescence data cannot directly provide well-constrained spore counts (Huffman et al., 2020). Other biological particles (bacteria and pollen) as well as certain types of non-biological particles can contribute to these measurements as well, although with varying fluorescence efficiency and as

a function of particle size and especially instrument operation and analysis procedures (Crawford et al., 2015; Perring et al., 2015; Savage et al., 2017; Toprak and Schnaiter, 2013). Further, weakly fluorescing spores can escape detection (Huffman et al., 2012). Therefore, we do not compare number concentrations directly, but focus on seasonal cycles and vertical profiles instead, for which we show time series and profiles that are normalized to their maximum value.

Few observational studies exist that cover a full seasonal cycle or longer. Two of the available datasets were collected in

Europe, and thus provide particularly valuable validation of our models beyond the domain for which they have been developed. The datasets used in this comparison are from a semi-rural site in Karlsruhe, Germany, where a WIBS-4 instrument was employed (Toprak and Schnaiter, 2013), from a boreal forest in Hyytiälä, Finland, and from a pine forest in Colorado, USA (Schumacher et al., 2013), where UV-APS instruments were used. Based on one distinct mode in their FBAP observations, Toprak and Schnaiter (2013) attributed their observations to a site-specific spore type. For the Hyytiälä site,

Manninen et al. (2014) suggest that fungal spores strongly contribute to PBAP numbers, based on spore counts. No dominant contributor to the FBAP concentrations has been identified at the Colorado site.

Our focus is on the normalized seasonal cycle, but we note that when comparing the absolute concentrations, we find a systematic low bias for the population and the statistical model and a high bias for the HS09 model. There a several reasons





why a low bias in the model simulations is reasonable. First, as previously noted, the FBAP concentrations from the WIBS and UV-APS instruments do not only consist of spores, but may contain bacteria and pollen (fragments) too, as well as interferences from non-biological particles. Second, at the Hyytiälä and Colorado sites, the instrument inlets were situated inside the canopy, where concentrations of bioaerosols are usually higher than above the canopy due to proximity to sources

(Crawford et al., 2014; Gabey et al., 2010). GEOS-Chem, on the other hand, does not include a canopy model, so its results are representative of the lowest atmospheric layer above the canopy. Finally, as noted in Section 2.1, the spore count measurements at AAAAI sites, which are used here to constrain the emissions used in GEOS-Chem, are a lower limit given the size limits of the sampling. Unfortunately, there are no co-located fluorescence and spore count measurements that can be compared directly to explore these differences.

For the normalized seasonal cycle, we find similar results for all three sites (plotted as 20-day rolling means in Fig. 9). Note that we correct for the fact that the emitting land fraction of the GEOS-Chem grid box over the Hyytiälä site is smaller than one, and that we exclude the period during which the ground surface was covered with snow at this site from the statistics, since this inhibits spore emission (Schumacher et al., 2013). For the Karlsruhe site, all model simulations capture the broad features of the seasonal cycles well, with low concentrations in winter (January to March), rising concentrations in spring and

peak concentrations in summer and fall (until October). The HS09 model, however, peaks too early in June, when the observations suggest only a minor increase, while the population model does not capture the rapid increase of concentrations in May and June. At Hyytiälä, the HS09 model shows a peak in July, which is not present in the observations or the other models, and the population model also misses the peak in early summer here. For both sites, all models show similar skill in capturing the seasonal variability. Only for the site in Germany, the population model captures the seasonal variability

somewhat better than the other models.

At the site in Colorado, all models have difficulty capturing the average behavior shown in Figure 9. The seasonal cycle at this site is composed of observations that span two calendar years, July 2011 to June 2012, which explains the sudden shift from high to low normalized concentrations in summer. For the period from January to July, all three models capture the concentration increase, with low concentrations from January to April, and a sharp increase from May onwards. Especially the

statistical model reproduces the timing and relative magnitude of this growth well. During the period from September through November, however, the statistical and population models fail to capture the relatively low spore concentrations. Only in December, all models capture the minimum in the concentrations that is present in the observations as well.

In addition, the agreement between model and measurement strongly depends on the choice of the temperature threshold below which emissions are shut off for the statistical and the population model. In Section 3.1, we set this threshold to 0°C, and here

we evaluate the effect of setting no temperature threshold and a threshold of 5°C, respectively. Figure S3 shows that setting no temperature threshold strongly degrades the model-measurement agreement, especially at the Karlsruhe site in December when modeled concentrations peak while FBAP concentrations actually have a minimum. For Hyytiälä, the model-measurement agreement decreases as well, but less than at the Karlsruhe site, because the period with snow cover was already excluded. On the other hand, when we set the threshold to 5°C, both the statistical and the population model reproduce the





seasonal cycles at both sites well, with $r^2$ between 0.62 and 0.71. The fact that this relatively arbitrary choice makes such a big difference for the ability of the models to reproduce the observed seasonal cycle suggests that the low availability of AAAAI observations during winter severely limits the derivation of emission schemes from those data.

In conclusion, calculated spore concentrations from the population and the statistical model capture the seasonal variations in FBAP concentrations with comparable skill as the HS09 model, although assumptions on the temperature threshold below which no emissions occur have a large influence on the performance of the former two models.

Vertical profiles of FBAP are available for several campaigns over the continental United States, including SEAC4RS (Ziemba et al., 2016) and IDEAS (Twohy et al., 2016), and over the North Atlantic from the NAAMES 2015, 2016 and 2017 campaigns (Behrenfeld et al., 2019). The SEAC4RS and IDEAS campaigns enable us to evaluate how well the model captures the vertical transport of fungal spore-like fluorescent particles close to source and the NAAMES campaigns characterize the transport of spores through continental outflow toward the North Atlantic. The North Atlantic Aerosol and Marine Ecosystems Study (NAAMES) included aerosol measurements from the NASA Wallops Flight Facility (WFF) C-130 based in St. John's, Newfoundland, Canada. Flight campaigns occurred in the fall of 2015 (9-Nov through 23-Nov), late-spring of 2016 (18-May through 1-June), and late-summer of 2017 (28-Aug through 19-Sept). The WIBS sampled iso-kinetically through a shrouded solid-diffuser inlet that efficiently samples particles up to 5µm aerodynamic diameter (McNaughton et al., 2007). WIBS was operated at a constant sample flowrate and concentrations were corrected to standard temperature and pressure (Ziemba et al., 2016). To exclude the possible influence of biomass burning, which can produce fluorescent aerosol (Savage et al., 2017), only the observations for which simultaneous acetonitrile concentrations are below 200 ppt were used. Cloud contaminated samples have been removed using coincident measurements from a set of wing-mounted optical probes.

Because of the same issues with the interpretation of FBAP measurements as mentioned above, we compare mean observed and simulated normalized vertical profiles for each campaign. When we compare the simulated concentrations with the observed profiles, we see that simulated normalized concentrations from GEOS-Chem generally agree well with the observed concentrations from the SEAC4RS and IDEAS flights (Fig. 10). For the SEAC4RS flights, all models capture the observed vertical profile. Potential temperature profiles agree well between model and observations, which gives confidence in the correct representation of convective transport by the model. For the IDEAS flights, the model slightly overestimates normalized concentrations around 650hPa and underestimates them between 600 and 500 hPa, but these difference fall within the variability in the observations. Overall, the model appears to generally capture the vertical transport of spores over their source regions. The dilution factor between BL and FT from these modeled profiles is about 0.3 for the SEAC4RS and about 0.6 for the IDEAS campaign (in Section 4 we explore how use of these dilution factors would impact our emissions derivation). For the campaigns over the North Atlantic, the model simulations underestimate the absolute concentrations (which are small; $<10$ L$^{-1}$) for all years and emissions schemes, with the exception of the HS09 scheme for the 2017 campaign. All years show concentration maxima between the 800 and 600 hPa levels (Fig. 10), which are the result of continental outflow of fluorescent particles. The simulations generally do not capture these relative profiles, and show decreasing concentrations with height,





with the exception of 2015, when all simulations reproduce the lower-tropospheric peak between 650 and 850 hPa, and 2017, when model simulations for the HS09 scheme peak at that same level. Given that the model captures the potential temperature profile for the 2016 campaign, it seems unlikely that local convective transport is the reason for the mismatch. Rather, it suggests that long-range transport of fungal spores and processing through continental outflow may not be well represented

by the model. This points to the need for further investigation of the transport and solubility of fungal spores.

## 4. Discussion and conclusions

We have developed new emission schemes for fungal spores for inclusion in regional and global models, based on a previously unexplored dataset of fungal spore counts at 66 locations across the United States. First, we calculated fungal spore emissions from observed concentrations by applying the boundary layer equilibrium assumption, yielding annual average fungal spore

emissions over all stations of $62\pm31$ m$^{-2}$ s$^{-1}$. Then, we developed two schemes to simulate the emissions of fungal spores at seasonal timescales over a wide range of land use types: a population model that simulates the growth of fungi and the production of spores and their emissions as a function of temperature, LAI and friction velocity, and a statistical model that relates spore emissions to meteorological and land surface drivers. The population model shows better skill at reproducing the seasonal cycle in the emissions than the statistical model, whereas both outperform the HS09 scheme.

After implementation in GEOS-Chem, we used the new schemes to calculate global emissions and burden of fungal spores. For the population and the statistical model, we estimate emissions of 3.4 and 3.7 Tg year$^{-1}$, respectively, both of which are substantially lower than the estimate of 31 Tg year$^{-1}$, generated by the HS09 scheme. These differences are largely the result of different assumptions about size, and the use of different observational constraints (fungal spore counts in this work, versus mannitol concentrations in HS09). This suggests that fungal spores contribute less to the organic aerosol budget of the

atmosphere and are likely less important for cloud and precipitation formation than previously estimated in models.
However, these numbers are sensitive to our assumptions on 1) the derivation of fluxes from concentrations, 2) emission model formulation and 3) transport and removal processes in the GEOS-Chem chemical transport model. Regarding the former, we assumed a dilution factor of 0.1 between the BL and FT that was derived from a few observations only. Lower assumed values do not have a significant impact on the calculated fluxes, as such a low dilution already yields upper limit estimates for the

calculated emissions. We can also estimate the dilution factor inherent to our GEOS-Chem simulations, by comparing BL and FT fungal spore concentrations over land, and find that this value is typically ~0.3. Using this value in our derivation of emissions would decrease the average calculated flux to $49\pm25$ m$^{-2}$ s$^{-1}$, which translates to 21 and 21% lower global emissions for the population and statistical model, respectively (Table S1). This analysis shows that uncertainties in the dilution factor directly impact the modeled emission fluxes, but do not change our finding that these fluxes are an order of magnitude or more

lower than those estimated in previous studies. Large uncertainties also remain on the efficiency of wet removal, since both the representation of precipitation and the formulation of wet deposition schemes are complex issues for global models. We test the sensitivity of the modeled fungal spore burden to wet deposition by changing the rain out efficiency from 1 to 0. This

change from full to no solubility has a large effect on the global burden (leading to an increase of 28 and 31 % for the population and the statistical model, respectively, when spores are assumed non-soluble; Table S1), but it has little effect on the normalized vertical profiles. This suggests that current observations are insufficient to constrain the solubility of spores in the model.

Limited validation of our model results is possible with datasets outside the US domain. For two European sites, we find that

the population model and the statistical model reproduce the seasonal cycle in FBAP concentrations with comparable skill to the HS09 model, although poor constraints on emissions in winter prohibit more definitive conclusions. A comparison with vertical FBAP profiles shows that normalized concentration profiles are represented well over source areas, but that the continental outflow of bioaerosols over the North Atlantic is not captured well by our model, suggesting a need to further investigate the transport and removal of fungal spores. Uncertainties in the spore count data which form the basis for the

emission schemes and in the attribution of fluorescent measurements to spore concentrations prohibit a more quantitative evaluation of the modeled spore concentrations.

Although our new emission schemes are based on the largest available database of spore counts, there remain considerable uncertainties in our characterization of the fungal spore bioaerosol budget. Additional efforts are needed to improve our understanding of the impacts of fungal spores on atmospheric processes. First, more flux measurements of fungal spores over

forests and other ecosystems would be very valuable to quantitatively evaluate the magnitude of the flux of spores into the atmosphere. Further, there is a critical need for long-term concentrations measurements for locations that are not included in the AAAAI dataset, particularly in areas with high simulated fluxes, such as Southeast Asia, and in ecosystems such as tropical forests, for which currently very little data is available. Further improvements in FBAP measurements to be able to more confidently extract fungal spore concentrations for further comparison would be useful. Finally, our analysis points out that

there remain critical gaps in our understanding of long-range transport of spores, which calls for further research efforts in convective transport, cloud processing and wet removal of fungal spores.

**Code and data availability**

The GEOS-Chem model code is available at http://acmg.seas.harvard.edu/geos/ (last access: 7 June 2020). The spore count

data are available from the AAAAI upon request. FBAP data are available from the references as cited in the text.

**Author contributions**

RJ and CLH designed the study. RJ performed the data analysis and the model simulations. ALS, AEP, JAH, ESR, CHT and LDZ provided spore count data and FBAP measurements used in the analysis. RJ and CLH wrote the paper with input from

the co-authors.

**Competing interests**

The authors declare that they have no conflict of interest.





## Acknowledgements

We thank Will Porter for discussion of the regression analysis, and Martin Schnaiter and Emre Toprak for sharing the Karlsruhe WIBS data. We gratefully acknowledge the use of the American Academy of Allergy, Asthma and Immunology (AAAAI) spore count data. This study was supported by the National Science Foundation (AGS-1564495). AEP was supported by the

NOAA Atmospheric Composition and Climate Program and the NOAA Health of the Atmosphere Program. JAH was supported by institutional funding from the University of Denver and the Max Planck Institute for Chemistry (MPIC). Colorado and Finland data were collected with support from the MPIC and the Mainz Bioaerosol Laboratory (MBAL), with special thanks to Ulrich Pöschl and Christopher Pöhlker. SEAC4RS measurements were supported by NASA's Upper Atmosphere Research Program, Radiation Sciences Program, and Tropospheric Chemistry Program. NAAMES measurements were

supported by the NASA Earth Venture Suborbital Program.

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





| Parameter | Fitted value | Unit |
|---|---|---|
| $b_0$ | $2.63 \times 10^{-5}$ | $m^2\ s^{-1}$ |
| $b_1$ | $6.10 \times 10^3$ | - |
| $b_2$ | 46.7 | - |
| $b_3$ | 59.0 | m |

**Table 1: Fitted parameters of the statistical model**

| Parameter | Fitted value | Allowed range | Description | Unit |
|---|---|---|---|---|
| $r_{max}$ | $7.81 \times 10^{-1}$ | 0-10 | Maximum growth rate | $day^{-1}$ |
| m | $1.42 \times 10^{-2}$ | >0 | Mortality rate | $day^{-1}$ |
| $T_{opt}$ | 27.5 | 0-35 | Optimum temperature for fungal growth | °C |
| $T_{max}$ | 31.4 | 10-40 | Maximum temperature for fungal growth | °C |
| $T_{min}$ | 0.0 | 0-20 | Minimum temperature for fungal growth | °C |
| $l_1$ | 72.0 | >0 | Parameter for LAI dependence | - |
| $l_2$ | 18.9 | >0 | Parameter for LAI dependence | - |
| $s_1$ | 10.6 | >0 | Parameter for u* dependence | $s\ m^{-1}$ |
| $s_2$ | $1.99 \times 10^{-2}$ | 0-1 | Parameter for u* dependence | $m\ s^{-1}$ |

**Table 2: Fitted parameters of the population model**

| Emission scheme | Emission (Tg year-1) | Burden (Gg) | Lifetime (days) |
|---|---|---|---|
| **Population model** | 3.4 | 20.0 | 2.1 |
| **Statistical model** | 3.7 | 15.3 | 1.4 |
| **HS09** | 31 | 130 | 1.1-2.6 |

**Table. 3: global emissions, burden and lifetime for fungal spores using the three different emission schemes**



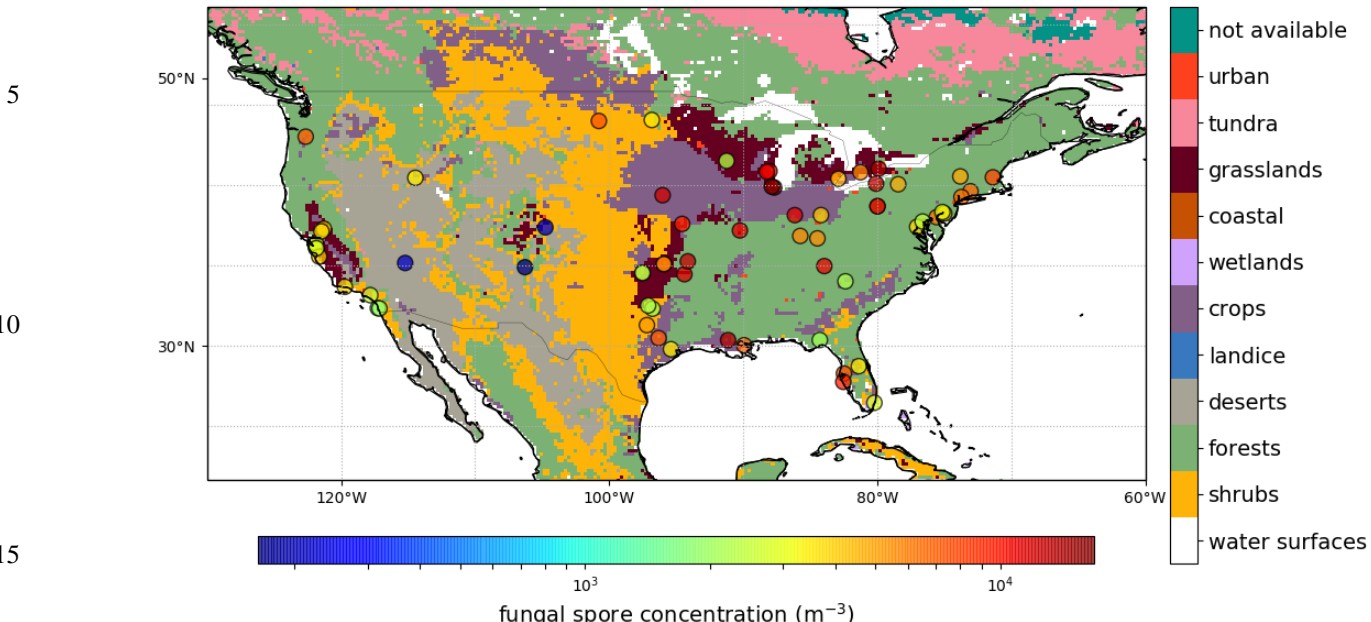

**Figure 1: Average observed fungal spore concentrations over the period 2003-2008 for all AAAAI stations (circles) shown on top of lumped land use classes bases on the Olson World Ecosystems (Olson, 2001)**





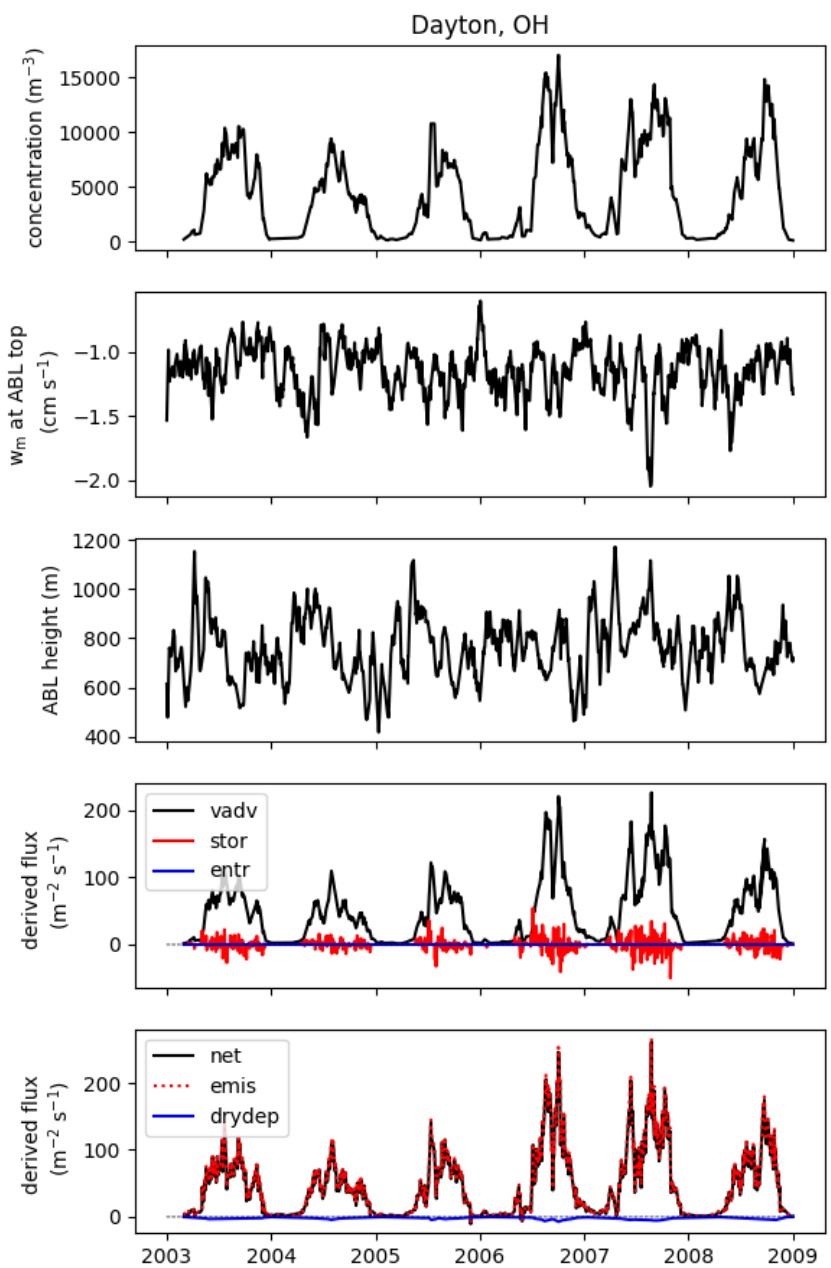

**Figure 2: Example of how emissions flux is derived from observed concentrations at one AAAAI site located in Dayton, OH. Shown here are 20-day running mean time series of a) fungal spore concentration, b) subsidence velocity at atmospheric boundary layer (ABL) top, c) maximum daily boundary layer height, d) contributions of vertical advection (vadv), storage (stor) and entrainment (entr) terms to the calculated flux, and e) calculated net flux, emission flux and dry deposition flux**



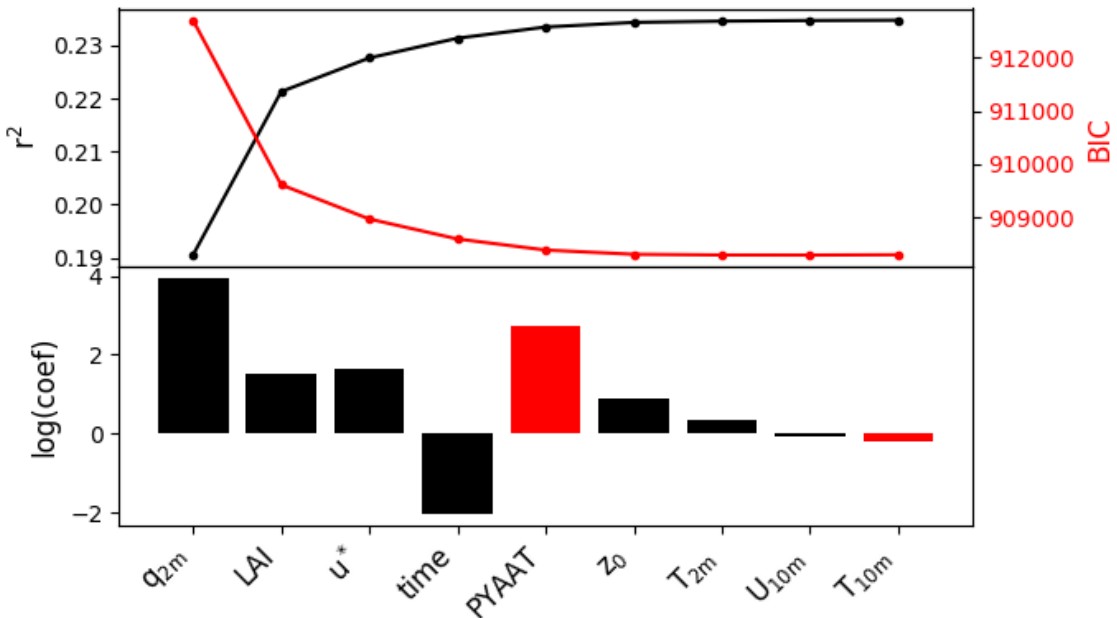

**Figure 3: results of the multivariate regression analysis: r2 and BIC after including each variable in the MLR analysis (top) and the logarithm of the regression coefficient for each variable (bottom). Red bars indicate a negative regression coefficient. Included variables are: specific humidity at 2 m ($q_{2m}$), leaf area index (LAI), friction velocity ($u^*$), time (expressed as number of days since start of time series), previous year annual average temperature (PYAAT), roughness length ($z_0$), temperature at 2 m ($T_{2m}$), wind speed at 10 m ($U_{10m}$), and temperature at 10 m ($T_{10m}$)**

**Figure 4: Model fungal spore emissions for the a) statistical scheme b) population scheme and c) HS09 scheme compared to the 20-day running derived-emission flux at all 66 AAAAI stations. Left panels show time series comparisons over 6 years; shaded areas show the standard deviation of the derived fluxes. Right panels show point-by-point comparison with fit parameters shown inset and the 1:1 line shown as a dashed line.**





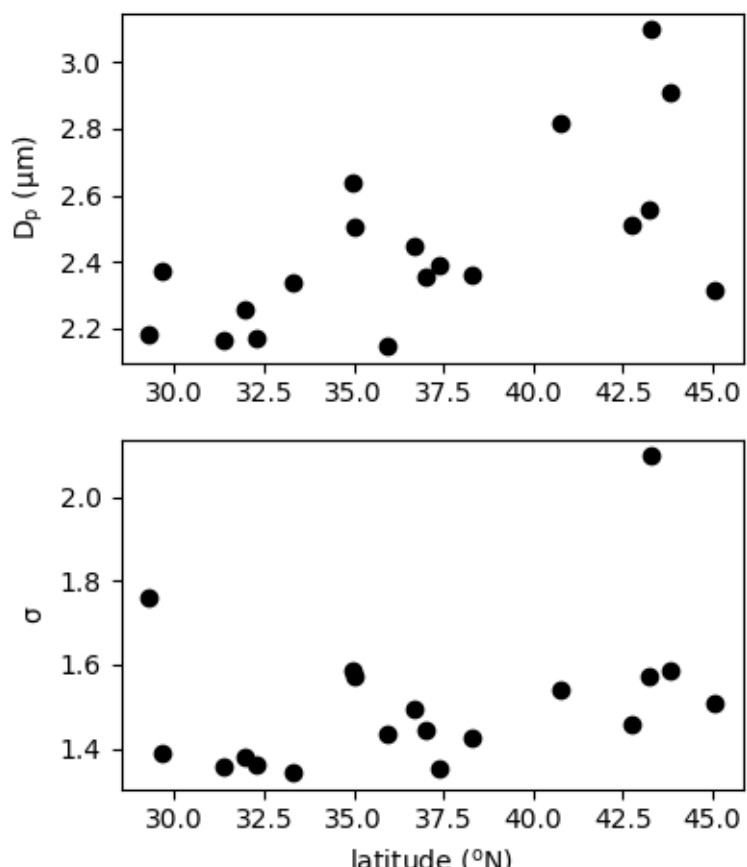

**Figure 5: Geometric mean diameter ($D_p$) and geometric standard deviation ($\sigma$) as a function of latitude for the number distribution of FBAP particles observed by WIBS over the continental US in 2016**





**Figure 6: Annual average fungal spore emissions and burden from the statistical (top), the population (center), and the HS09 model (bottom)**



**Figure 7: Seasonal cycles of fungal spore emissions (left) and concentrations (right) for the statistical, the population and the HS09 model. Note the different scales on the y-axis**





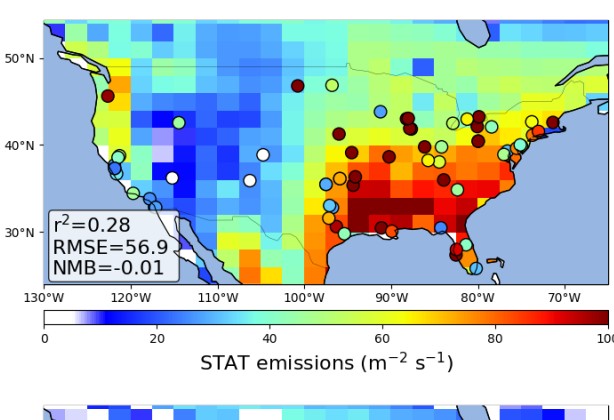

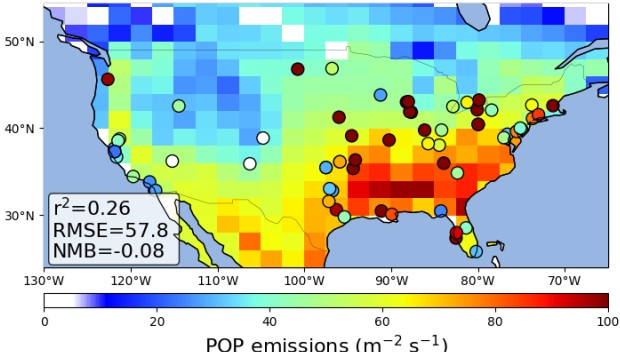

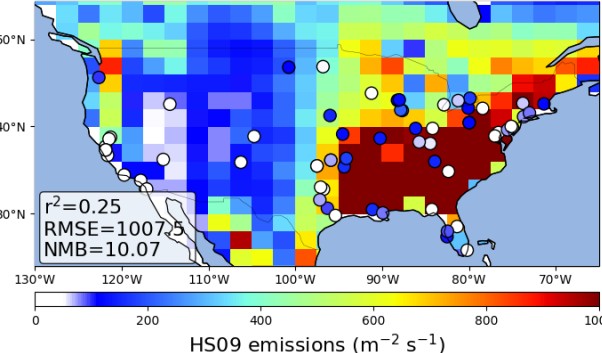




**Figure 8: Comparison of simulated to calculated emission fluxes of fungal spores for the statistical model (top), the population model (middle) and the HS09 model (bottom). Note the different scale for the bottom figure. Statistics describing the comparisons are shown inset.**



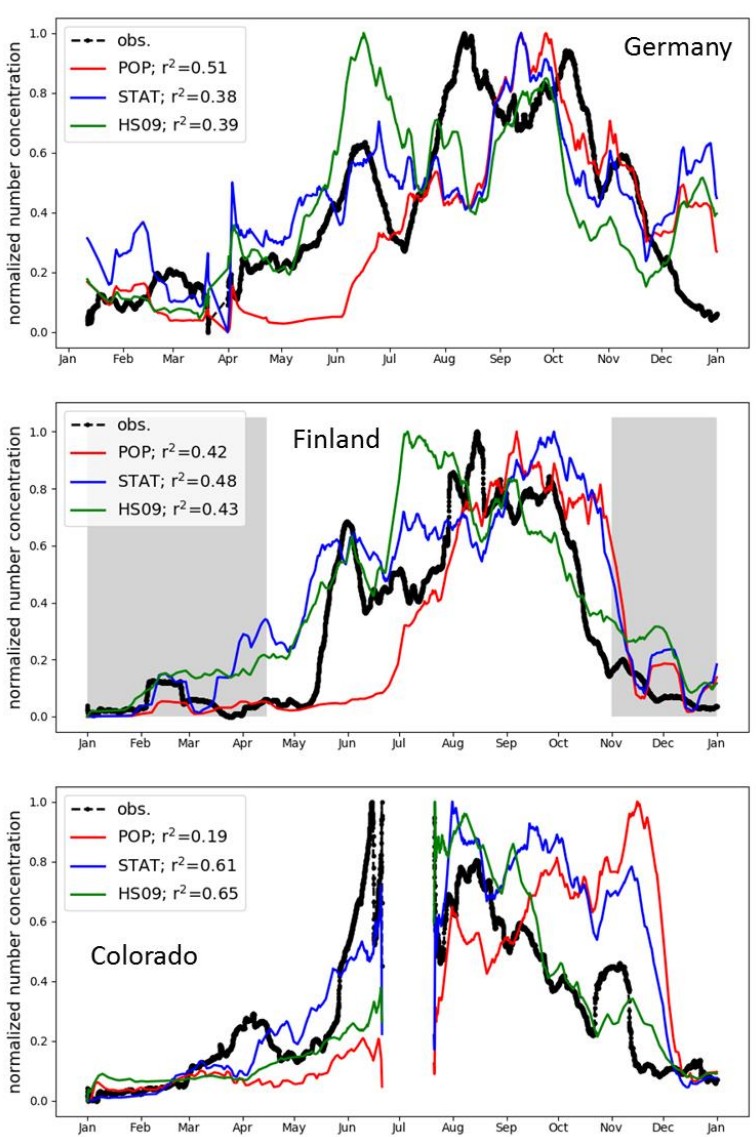

**Figure 9: Normalized seasonal cycle in fungal spore and FBAP concentrations in Germany, Finland, and Colorado (see text for details). Observations (black) are compared to the simulated concentrations from the population model (red), the statistical model (blue) and the HS09 scheme (green). The shaded area indicates periods with snow cover; statistics are given for snow free period only.**







**Figure 10: Normalized vertical profiles of fluorescent biological aerosol particles (FBAP) from 5 campaigns (black) compared with normalized vertical profiles of fungal spore number concentrations from 3 model simulations: the population model (blue), the statistical model (red) and the HS09 model (green). Standard deviation of observations in each 50 hPa pressure bin are shown in grey. The upper right panel shows the flight tracks for each campaign.**

