# Peer review of "Drivers of the fungal spore bioaerosol budget: observational analysis and global modelling"

_Atmospheric Chemistry and Physics, 2020_

## Referee Comment (RC1) · Anonymous Referee #1 · 9 Aug 2020

Summary

This study presents two new schemes of fungal spore emissions and compares model results with the well-established parameterization of Heald and Spracklen (2009), as well as, with available observations. This work concludes that the new and more sophisticated emission schemes produce about one order of magnitude lower emissions than previously estimated. I find this paper well written and the conclusions very useful in exploring the uncertainties of different fungal spore emission schemes, along with the calculated atmospheric burden by global models. Some minor issues, however, can be addressed by the authors before the final publication in ACP, to help the reader to better understand the proposed parameterizations.

General comments

[Figure]

The authors state that the new parameterizations result in emission strengths of about an order of magnitude lower than the HS09 model and thus, fungal spores contribute less to the total organic aerosol burden in the atmosphere. However, the HS09 model presents results for both "PM2.5" (diameter < 2.5 $\mu$m) and "PM10" (2.5 $\mu$m < diameter < 10 $\mu$m) fungal spores, in contrast to the emission schemes of this work that are based only on spores with a diameter of 2.5 $\mu$m ($\sigma$ = 1.5). Considering that the emission schemes are highly sensitive to the assumed size of the spores, I wonder whether such a comparison is fair by only referring to the total emitted masses and not also to the respective particle sizes. Further discussion is needed to support this conclusion since the size distribution(s) of the compared emission schemes (i.e., new vs. old) significantly differ.

Specific comments

1. The authors present the new fungal spores' emission schemes in Sect. 2. Although the HS09 parameterization is well established, a somewhat more extended discussion of that parameterization would be useful for the reader (a short discussion is, nevertheless, presented in Sects. 2.5 and 4.). For example, the authors could discuss more on the main differences between the old and the new schemes, i.e.: What is the main driver for the resulted overestimation of the previous scheme compared to the new schemes? Is it only the observations used (i.e., spores counts vs. mannitol concentrations) or/and the sizes of the observed fungal spores? Do the current parameterizations use more advanced statistical tools than previously? And possibly, what global emissions would have been derived if the authors had used mannitol concentrations, as in Heald and Spracklen (2009), on the spores considered in this study? Some of these issues were touched in the discussion section, but a more detailed analysis would be helpful.

2. Page 11, lines 7-12: The dry deposition and the sedimentation budget terms of the model should be more explicitly presented and discussed. Heald and Spracklen (2009) used two modes to parameterize the fungal spore emissions, i.e., a fine mode,

with a diameter < 2.5 $\mu$m, and a coarse mode, with a diameter < 10 $\mu$m. Although here the authors assume a fungal spore diameter of 2.5$\mu$m ($\sigma$=1.5), they also assume that the spores are present only in the coarse mode. Do the authors use a different size distribution scheme for this work compared to HS09? How different is this assumption with the one used by Heald and Spracklen (2009)? How do they compare? A more detailed discussion of the aerosol size distribution scheme(s) of the model is necessary to understand these differences.

3. Page 12, line 8: What molecular weight is used in the model for the fungal spores?

4. Page 12, lines 14-16: Considering that for the HS09 model, the fungal spores are present mostly in the coarse mode, sedimentation should be a significant process for their atmospheric lifetime calculation. For this, the respective annual budgets (ideally for all model simulations of this paper) should be presented in Table 3 and discussed in more detail in the manuscript. Besides, an additional Table with all simulations performed for this study would be also very useful.

5. Page 13, lines 25-26: Do the authors refer here to a new simulation (i.e., with fully emitted land-cover)? If yes, what is the impact on the calculated fungal spores' emissions and burdens? How much would that differ compared to the standard simulation? Is this correction applied only to specific boxes (i.e., those include the coordinates of the observation sites used in this work for model evaluation)?

6. Page 17, lines 4-5 & Page 18, lines 1-3: The authors discuss the impact of fungal spores' solubility assumptions (i.e., insoluble vs. fully soluble) on the long-range transport and global burden. Considering, however, a mean lifetime for all simulation of up to ~2 days in the model, the conversion from insoluble to soluble via atmospheric processes (e.g., assuming 1.2-day e-folding conversion from hydrophobic to hydrophilic) may be potentially significant. A short discussion on fungal spores' aging would be here useful.

7. Page 17, line 19: "This suggests that fungal spores contribute less to the organic

aerosol budget. . ." Is this statement valid only for PM2.5 spores, or also for spores up to PM10, when the respective emission schemes are applied?

Technical corrections

i. Page 3, lines 23-28: A more detailed outlook paragraph at the end of Sect. 1 would be useful for the reader.

ii. Page 33: Figure 7 fits better in the supplement.

iii. Page 34: Please explain better in the caption the "simulated" vs. "calculated" emission fluxes.

iv. Page 35: Please explain better how the "normalized" vertical profiles are calculated.

---

## Referee Comment (RC2) · Anonymous Referee #2 · 20 Aug 2020

Review of acp-2020-569

Drivers of the fungal spore bioaerosol budget: observational analysis and global modelling

The present study uses a multi-year spore count dataset in order to derive spore number emissions. Two different emission models (a statistical and a population model) are fitted to these derived emissions. Finally, the two emission models are included into the GEOS-Chem aerosol transport model. The resulting spore masses are compared to the spore emission scheme that was already implemented in GEOS-Chem. Large difference were found between the old and the two schemes developed in this study. The three spore emission schemes were evaluated against the dataset that was used to derive the emissions and against ground-level FBAP observations and vertical

FBAP profiles. In general, the method to derive spore emissions is scientifically sound and promising for to give a more complete picture if more datasets become available to constrain the emission schemes. Furthermore, I really appreciate the study design involving a variety of different observations for model evaluation. However, in particular the sections discussing the GEOS-Chem results and evaluation (3.3. and 3.4) have some considerable weaknesses when it comes to give reasons and explanations for the findings and differences between the schemes and to give necessary information. For example it is not clear if the time period of the different observations was actually simulated or if the observations were compared to simulations of the year 2016 (and perhaps 2015), which is indicated by section 3.1. At least, I would need more information in order to evaluate the findings of this study. My concerns and questions are addressed in the comments below.

Major comments p.4, l.8: How does the sampling rate of 3 days a week relate to the filtering by rain? How long is the collection time in these cases? From the description, I assume 2-3 days. Was the datapoint excluded if rain was found on any of the days of the collection?

p.5, l.6-7: Does this also involve rainfall in the surrounding grid cells?

p.6, l.13-16: I think, the method applied in this work is correct, however I don't understand the reasons given in these sentences. The description of the method is difficult to follow and could be extended by providing more information on certain assumptions. I can't follow the explanation here. Can the authors please explain why short atmospheric lifetimes necessarily lead to smaller concentration differences in the horizontal compared to the vertical? I would have argued that long lifetimes lead to small concentration differences due to longer time for mixing. Since horizontal wind speed is usually orders of magnitude stronger than vertical wind speed, already concentration differences smaller than the assumed here for the vertical (factor of 0.1) may result in the same order of magnitude horizontal advection. Having a read in Bakwin et al., 2004, one can find that they assume horizontal advection to be neglected due to the averaging period of 1month, which perhaps might be the case in regions with no prevailing winds and horizontal relatively homogeneously distributed sources. It's more the long averaging time that smoothes out horizontal differences. By neglecting horizontal advection, the authors implicitly assume long-term horizontal homogeneity, which is also the reason for why only subsidence velocity is applied. I think that this needs to be better described in the text.

p.12, l.32: "...are mainly caused by the occurrence of wet deposition". It sounds reasonable. However, can the authors please show the modelled wet deposition rates similar to Figure 7 or at least modelled rainfall in their answer to this comment? According to p.10, l.17-18, the population model has a "delayed response to temperature and LAI, due to the growth and mortality of the fungi". Can that also play a role here? How long is this delay?

p.13, l.5-7: I don't see the strong difference that is described here. Actually, it seems the July and August concentration of statistical and population model are quite similar (0.04-0.05 $\mu$g m-3).

p.13, l.14: Since wet and dry deposition are the only described loss processes (if I understand correctly), and both loss processes scale directly with the available concentration (i.e. the relative loss is independent of the actual concentration), this statement ("Due to the strong emissions of the HS09 model, emissions and concentrations have similar cycles.") seems wrong. Can the authors clarify what they meant to express?

p.13, l.16-17: Isn't that what the sentence before is saying for the HS09 model? So, I don't understand what different interplay between wet deposition and HS09 and wet deposition and the other two models the authors are targeting at here. For me, the different size assumptions, as explained in the sentence before, seem to be more the cause for different response to the same modelled rainfall rate. However, plotting modelled wet deposition rates would give proof to this speculation.

p.13, l.25-26: There are a number of stations close to the coast. For me, not necessarily, filling no-emitting water surface with emissions leads to a better comparison between model and observations, since also in the observations in terms of spores presumably clean marine air masses might lead to low concentrations. Furthermore, also the observational sites experience spores from probably a variety of land covers in their surrounding. Can the spore counts really be estimated to be dominated by local sources without a strong contribution of long-range transport (at least from around 200km)? Based on the lifetime shown in Table 3, long-range transport over this distance is occurring frequently. Can the authors please show a comparison of their "fully emitting land cover" simulation and the simulation using the original land cover at least for the stations whose land cover has been changed by this assumption? If I understood correctly, the original land cover was used for the results shown in e.g., Fig. 6.

p.13, l.26: Does "fully emitting land cover" also involve grid cells containing nothing but ocean? From the text, this is not clear.

p.13., l.27-29: I would have guessed that in particular the variability of meteorology in different years is one of the main causes of differences. Since different years were observed and simulated, how is the correlation coefficient calculated, i.e. which time-series is/are or population mean are compared against each other? On p. 14, l.5, low "skill in reproducing seasonal variations at the AAAAI stations" is determined by small correlation coefficients. Even for long-term averages such as seasons, the meteorological driving variables might be different in different years. Can the authors please show a comparison of the 2016 meteorological conditions and the observational years?

p.13, l.34: The 25% / 75% share between fine and coarse mode is reported for the mass in the abstract of Heald and Spracklen (2009). So, for spore mass, this statement absolutely makes sense. However, in Fig. 8, I think the number emission flux is shown, right? Does the HS09 scheme really show a larger number of coarse mode spores than number of fine mode spores? Heald and Spracklen (2009) actually state "...we obtain number concentrations of 10ˆ5 m-3 (fine) and 8 x 10ˆ3 m-3 (coarse)."

(p. 4, paragraph [9]). This does not seem to be inconsistent with the observed size distribution, which the authors refer to in the next sentence. Furthermore, it suggests that the overestimation is either general (i.e., both in fine and coarse mode) due to the applied method by Heald and Spracklen (2009) or at least in the fine mode. Can the authors please clarify what they meant to express here?

Section 3.4: I assume that the observed FBAP concentrations are from other years than 2016. Are the same years simulated or is the 2016 simulation used instead? This is not stated in the text and should be mentioned. In section 3.1 it is written that only the years 2015 and 2016 were simulated. If other simulated time periods than the observed ones were used, the knowledge gain from the results presented in this section is limited. I would not necessarily expect good comparison for single peaks in the seasonal cycle and timeseries statistics as different meteorological conditions and timing can vary greatly between different years. The section should include clear statements on these limitations if other years were simulated than observed. In particular it should then be checked how the meteorological variables that drive the emissions compare at or close to the sites between the different years.

p.15, l.15-16: Can the authors clarify what is meant here? For me, it looks like that at Karlsuhe (which I believe the statement is referring to) HS09 and observations peak right at the same time in June, however HS09 is just overestimating. In case you mean that the maximum concentration in HS09 is in June whereas it is Aug-Oct in the observation, I recommend to revise this sentence.

p.16, Section on the comparison to vertical profiles of FBAP: Were the same time periods simulated in which the obersvations took place?

p.16, l.15-16: Which modelled time period was compared to these observations?

p.17, l.2-3: How does this compare for the other years with observations (2015 and 2017)? At least, 2015 was simulated.

p.17, l.17-18: "These differences are largely the result of different assumptions about size...". If I understood correctly, the HS09 is emitting the mass. So size does not play a dominant role for the mass concentration. Is the HS09 emission in GEOS-Chem implemented as mass- or number-based emission scheme?

Other comments p.4, l.10: What does "no local spore or pollen sources" mean? How close is local in this context?

p.5, l.17: Do you mean subsidence of cold air? If not, then please further explain what is meant here.

p.6, l.4: The abbreviations BL and FT have not been introduced, yet.

p.6, l.7: I'm not sure if the abbreviation for local time is well known.

p.6, l.9-10: I assume that the boundary layer height is also taken from NARR data? Further, the authors write "mean height of the afternoon boundary layer", however, in the Figure caption of Fig. 2 they write "c) maximum daily boundary layer height". Can the authors please clarify, which was used?

p.6, l.10: Double "daytime" in "the daytime mixed-layer during daytime".

p.6, l.33-34: How was this calculated?

p.8, l.23-25: This sentence is rather complicated. It could perhaps be split into two sentences for easier reading?

p.8, l.31: I think, referring to section 2.5 where the fitted parameters are presented or even to Table 2 already in the beginning of section 2.4 is beneficial for the reader here.

p.9, l.22: Perhaps nothing to bother before type setting, but the font seems to have changed.

p.10., l.17-18: Double "in the population model".

p.10., l.25: Perhaps name it "derived spore emissions" to better make clear that the

observed emissions are meant.

p.11, l.5: Can the authors please show the comparison of LAI for 2008 and 2016 in an answer to this comment?

p.12, l.10: For me, burden is usually a column quantity or mass / number in the whole atmosphere. However, shown in Figure 7 is a concentration [$\mu$g m-3] according to the axis label. I therefore recommend to use "concentrations" instead of "burden", as it is done later in this section.

p.13, l.18-19: This sentence reads a bit difficult. First, at this point, it is not clear (since it is the beginning of a new paragraph) that with "both schemes" the authors mean statistical and population model. Second, the fragment "as well as the HS09 scheme" is badly placed / written for easy understanding.

p.13, l.19-20: I get your point, but written this way, it seems like a rather strong statement. I would have not assumed it, especially since a different time period is simulated (2016) than observed (2003-2008).

p.13, l.30: The reference to Figure 8 should already be given somewhere in the beginning of this paragraph.

p.14, l.6: Just for my understanding. The comparison and correlation coefficients in Fig. 4 are for the timeseries of the emission mean of all stations? If so, I suggest to write this clearer in the respective section.

p.15, l.24: Actually, the population model stays rather low. I would not call the increase "sharp" for this particular model.

p.15, l.29: Why was the temperature threshold not introduced for the HS09 model?

p.18, l.8: Please consider revising this statement ("continental outflow of bioaerosols"). Only spores were simulated.

Table 1: Assuming [q] = g/g (or similar) and [LAI] = m2m-2 the unit of b1 and b2 should

be m2s-1?

Table 3: Typo: "Table. 3" -> "Table 3".

Figure and Table captions: Probably not something to bother at this stage: Sometimes dots are missing at the end and non-capital letters in the beginning of Figure and Table captions, in both the manuscript and the supplement.

Figure 2: a-e missing in the Figure itself, but it is referred to in the text and the figure caption.

Figure 3: In the Figure caption, "2" is not in superscript in rˆ2.

Figure 4: Perhaps typo in the figure caption (I'm not a native speaker)? "20-day running derived-emission flux" -> "20-day running mean derived emission flux".

Figure 4: "fit parameters shown inset". This is not the case. However, it is not needed since fit parameters are presented in Table 1 and 2 very nicely.

---

## Referee Comment (RC3) · Anonymous Referee #3 · 9 Sep 2020

Overall comments:

This manuscript presents the development of a new parameterization, suitable for use in regional and global atmospheric models, of the emissions of fungal spores to the atmosphere, as a function of meteorological and land surface parameters. The new parameterization is derived based on a large dataset of fungal spore counts from the American Academy of Allergy, Asthma and Immunology (AAAAI), which has previously not been exploited for this purpose. Since visual counts of fungal spores are widely understood to be the most reliable measurement of atmospheric fungal spore concentrations that is typically available (despite potential limitations), a parameterization based on this new data source can be expected to have greater reliability than previous parameterizations based on other proxy measurements (e.g., mannitol con-

centrations, and concentrations of fluorescent biological aerosol particles, FBAP). The new parameterization should especially be relevant within the region from which the observational data were obtained (North America), but has been developed on the basis of variables that are globally available from observational datasets and/or within atmospheric models.

In addition, the new parameterization uses an approach to estimating the relationship between fluxes and near-surface concentrations based in a simplified approach to modelling the convective boundary layer that involves some limitations and assumptions, but which is more sophisticated than (and likely an improvement upon) the approaches taken in the development of some earlier parameterizations for fungal spore emissions. The parameterization is selected via a regression model, which is similar to the approach taken in Heald and Spracklen (2009), but which considers more variables and uses an improved statistical approach for model selection (i.e. multiple linear regression with model selection via the Bayesian information criterion to select the best model while avoiding over-fitting). Also, a biological-growth-based model is proposed in addition to the statistical regression model. Finally, the new parameterization is evaluated by comparison with normalized FBAP measurements (seasonal cycles and vertical profiles), and several sensitivities of the model are discussed.

In summary, this paper represents a significant advance in emissions modelling of fungal spores, and is within the scope of Atmospheric Chemistry and Physics. Most of the questions I had are already addressed by the authors with appropriate caveats in the manuscript in its current form. The neglect of horizontal advection in the inference of emission fluxes is likely a meaningful limitation, but one that is not possible to address with the approach/framework used here. Diurnal cycles of emissions (and their interaction with the diurnal cycle of the convective boundary layer) are also not addressed, but it appear that the existing data do not have sufficient time resolution to allow investigation of these cycles.

Based on my evaluation, I recommend that it be published after the following questions
and comments are addressed.

General questions and comments:

1. The main question I had about this paper is regarding the equilibrium boundary layer approach used to derive the flux estimates. I was not entirely convinced that the prior use of this method for inferring $CO_2$ fluxes is adequate justification for its use in inferring aerosol fluxes, since $CO_2$ is considerably more well-mixed in the atmosphere and has fewer complicating removal processes (especially wet removal). The study by Perring et al. (2015) is cited as showing that FBAP concentrations decline with altitude within the PBL, which seems to contradict the reliance on the assumption of well-mixedness. The approach relies on the assumption that convection maintains a well-mixed boundary layer; this assumption will not always be met, and there are likely systematic relationships between the times when the assumption is violated and some of the model's predictor variables (e.g., near-surface temperature). Diurnal cycles in emissions could also complicate the validity of the approach.

I think some discussion/analysis of how frequently the underlying assumptions of this approach are likely to hold would be warranted – especially the assumption of a boundary layer that is well-mixed with respect to both scalars and aerosols.

2. It strikes me as almost slightly contradictory that the temperature plays such a small role in the statistical model obtained via linear regression (Figure 3), yet the threshold value in temperature is shown to have a large impact on simulated emissions, and temperature also is a key variable in the population model. A priori, I would expect that fungal spore growth has an important, but non-linear, dependence on temperature, where growth would be inhibited at colder temperatures that are sub-optimal for fungal spore growth (as is also embodied in the population growth model). I wonder if the model would show a dependency on T if the analysis were repeated with a different statistical (or machine learning) method that allows for potential nonlinear dependencies. I recognize that would entail a significant amount of work (essentially repeating

the entire study), which is not necessary (and might not lead to improvement!).

But here I think it would be helpful if the authors could comment on (1) whether such approaches were tried and discarded for some reason, and (2) whether there is any notable relationship between the model-data mismatches (in modelled versus derived emissions and likely predictor variables including T at 2m and 10m (as might be revealed by a scatterplot).

3. The normalization of FBAP to compare with the spore data is appropriate considering the limitations of both types of observations. However, I think the normalization factors should be reported, as it would be informative for readers to know how much scaling had to be applied and how consistent or different this was between the datasets. Additionally, for the normalized vertical profiles in Figure 10, I was unable to find an explanation in the text of how the normalization factor was determined (I think so that the largest value in each vertical profile is 1?).

4. A key difference between the new proposed scheme and the HS09 scheme, which I think is not discussed, is the geographic representativeness. The mannitol data used in the HS09 scheme (Elbert et al., 2007; Table A3) includes a large number of data points from tropical rainforests of Brazil, which are not represented in the AAAAI dataset, as well as some extratropical data, which are mostly from Europe. It should be pointed out explicitly to readers that the geographic sampling is quite different from the data used for the previous parameterization (in addition to the differences in the measurement type and assumed size distribution, which are already noted).

Minor and typographical comments: P 6, l. 18-19 and l. 23-24 are partially redundant. p. 7, l. 24: some commas missing here inside the parentheses

---

## Author Comment (AC1) · 31 Dec 2020

In this document, we have included the comments of all 3 referees (in black) and our replies to these comments (in blue)

**Referee #1**

Summary
This study presents two new schemes of fungal spore emissions and compares model results with the well-established parameterization of Heald and Spracklen (2009), as well as, with available observations. This work concludes that the new and more sophisticated emission schemes produce about one order of magnitude lower emissions than previously estimated. I find this paper well written and the conclusions very useful in exploring the uncertainties of different fungal spore emission schemes, along with the calculated atmospheric burden by global models. Some minor issues, however, can be addressed by the authors before the final publication in ACP, to help the reader to better understand the proposed parameterizations.

We would like to thank Referee #1 for the constructive remarks on our manuscript. We will revise the MS according to these comments as described below:

General comments
The authors state that the new parameterizations result in emission strengths of about an order of magnitude lower than the HS09 model and thus, fungal spores contribute less to the total organic aerosol burden in the atmosphere. However, the HS09 model presents results for both "PM2.5" (diameter < 2.5 μm) and "PM10" (2.5 μm < diameter < 10 μm) fungal spores, in contrast to the emission schemes of this work that are based only on spores with a diameter of 2.5 μm (σ = 1.5). Considering that the emission schemes are highly sensitive to the assumed size of the spores, I wonder whether such a comparison is fair by only referring to the total emitted masses and not also to the respective particle sizes. Further discussion is needed to support this conclusion since the size distribution(s) of the compared emission schemes (i.e., new vs. old) significantly differ.

The different assumptions on the size distribution are the consequence of the datasets on which the emission parameterizations from the present study and HS09 are based: spore counts vs. sugar alcohol concentrations in particles, respectively. HS09 assumed that 20% of the emissions take place in the fine ('PM2.5') mode and the remaining 80% in the coarse ('PM10') mode, based on the mannitol measurements by Elbert et al. (2007) in fine and coarse mode particles, respectively. However, they did not have measured size distributions of spores available to further constrain those numbers. In contrast, we use spore counts as basis for the development of our emission scheme. Since the Burkard spore trap has no upper size limit (see section 2.1), we don't think that the spore counts miss a large part of coarse mode spores (if anything, we may miss a part of the smallest spores, as discussed in Section 2.1). Additionally, we use FBAP observations as constraint on the spore size distribution (Fig. 5), which we think is currently the most reliable observation available on the size distribution of spores in the atmosphere.
Therefore, although a direct comparison is hard due to the different data sources, we think that our constraints on emitted number and size distribution of spores are more robust than those that were available for HS09.

We will add in section 3.2 (p. 12, l. 7):
"Although a direct comparison is hard due to the different data sources, we think that these constraints on emitted number and size distribution of spores are more robust than those that were available for HS09."

Specific comments
1. The authors present the new fungal spores' emission schemes in Sect. 2. Although the HS09 parameterization is well established, a somewhat more extended discussion of that parameterization would be useful for the reader (a short discussion is, nevertheless, presented in Sects. 2.5 and 4.). For example, the authors could discuss more on the main differences between the old and the new schemes, i.e.: What is the main driver for the resulted overestimation of the previous scheme compared to the new schemes? Is it only the observations used (i.e., spores counts vs. mannitol concentrations) or/and the sizes of the observed fungal spores? Do the current parameterizations use more advanced statistical tools than previously? And possibly, what global emissions would have been derived if the authors had used

mannitol concentrations, as in Heald and Spracklen (2009), on the spores considered in this study? Some of these issues were touched in the discussion section, but a more detailed analysis would be helpful.

We break up this question in different parts, and answer the sub-questions below:

"What is the main driver for the resulted overestimation of the previous scheme compared to the new schemes? Is it only the observations used (i.e., spores counts vs. mannitol concentrations) or/and the sizes of the observed fungal spores?'

Drivers of differences in descending order of importance:
1. Presence of coarse mode in HS09, which contains 74% of the emitted mass in the HS09 scheme (but note that the fine mode from HS09 contains ~2 times more emitted mass than the two new schemes)
2. Assumptions on size distributions, but as in the reply to the general comment above, we believe that the size distribution as assumed for the new schemes is better constrained by observations than the fine and coarse modes in HS09. Note that the AAAAI data is a lower limit because the smaller spores are detected with less than 100% efficiency (see Section 2.1)
3. Locations of observations: HS09 used observations from tropical forests, which are expected to show higher concentrations of spores than temperate ecosystems as used in the present study. The absence of observations from tropical ecosystems is a limitation on the new parameterizations, so more spore count data from those ecosystems would be very valuable for evaluating the new schemes and/or to develop emission parameterizations for tropical ecosystems.

In Section 2.5 (around line 23), we will indicate that the mannitol observations (which are indirect constraint on spore counts) were taken from a handful of sites around the world and did not have the fully resolved seasonal cycle of the AAAAI observations, and thus these differences and uncertainties result in a factor of two difference when fitting HS09.

In Section 3.3, first paragraph (around line 20) we will insert the reasons for the differences in the global model simulations that are listed above. And we will move up the sentence "An overview…" on line 23-24 to be the second sentence of Section 3.3.

'Do the current parameterizations use more advanced statistical tools than previously?'

There are a few differences here:
1. In the current parameterizations there was a conversion from concentration to emission, before fitting the emission scheme
2. The statistical model uses somewhat more advanced statistical methods than HS09: multi-variate vs. simple linear regression. Moreover, the BIC is used as a measure of relative model performance, compared to the correlation coefficient in HS09
3. We here use a non-linear least-squares minimization algorithm to fit the stat and pop model to the observed spore counts, while HS09 used the reduced-major-axis method for fitting: however, we think that the choice of fitting method is not responsible for the large differences between the old and new schemes

'And possibly, what global emissions would have been derived if the authors had used mannitol concentrations, as in Heald and Spracklen (2009), on the spores considered in this study?'

It's not clear to us how to interpret this question. We could not have applied the same statistical techniques as we used to derive the new schemes to the mannitol data, because we don't have a time resolved full year of mannitol observations. However, it seems unlikely that this would have led to significantly different numbers for the global emissions than the HS09 scheme. Moreover, to apply the same method to those data would also require the conversion from mannitol concentrations to emissions first.
The other way around, we fitted the HS09 scheme to the AAAAI data (Section 2.5, Fig. 4), where it showed the least skill of the 3 emission schemes in reproducing the seasonal cycle in emissions. This fitting procedure also led to a coefficient which was about a factor 2 lower than the original value (p.10, l.22), which is consistent with the HS09 scheme predicting global emissions in the fine mode that are a factor of ~2 higher than the other schemes, when the original coefficient was applied in the GEOS-Chem simulations.

2. Page 11, lines 7-12: The dry deposition and the sedimentation budget terms of the model should be more explicitly presented and discussed. Heald and Spracklen (2009) used two modes to parameterize the fungal spore emissions, i.e., a fine mode, with a diameter < 2.5 µm, and a coarse mode, with a diameter < 10 µm. Although here the authors assume a fungal spore diameter of 2.5µm (σ=1.5), they also assume that the spores are present only in the coarse mode. Do the authors use a different size distribution scheme for this work compared to HS09? How different is this assumption with the one used by Heald and Spracklen (2009)? How do they compare? A more detailed discussion of the aerosol size distribution scheme(s) of the model is necessary to understand these differences.

Note that wet deposition is a more important removal mechanism in our simulations than dry deposition, with lifetimes against dry deposition of >50 days and lifetimes against wet deposition of 1.5-2 days. We will add in Table 3 the lifetime against dry and wet deposition separately.

*Table 3: global emissions, burden and lifetime for fungal spores using the three different emission schemes*

| Emission scheme | Emission (Tg year-1) | Burden (Gg) | Lifetime (days) | Lifetime dry dep. (days) | Lifetime wet dep. (days) |
|---|---|---|---|---|---|
| **Population model** | 3.4 | 20.0 | 2.1 | 54 | 1.5 |
| **Statistical model** | 3.7 | 15.3 | 1.4 | 64 | 2.1 |
| **HS09** | 31 | 130 | 1.1-2.6 | 21-48 | 1.1-2.7 |

In the GEOS-Chem wet deposition module, a distinction is made between fine and coarse mode aerosols, where coarse aerosol species are species with an effective radius >= 1um. Therefore, we have assigned the spores from the new emission schemes to the coarse mode in the wet deposition calculations. In the dry deposition calculations, the mean diameter of the assumed size distribution is applied. We will clarify this in the text.

'Do the authors use a different size distribution scheme for this work compared to HS09? How different is this assumption with the one used by Heald and Spracklen (2009)? How do they compare?'

For the simulations with the HS09 scheme, we have assumed monodisperse size distributions with Dp =1.25 µm and Dp = 6.25 µm for the fine and coarse modes, respectively, which is the same as in that study.
We will add on p. 11 line 25: "Based on mannitol observations in both the fine and coarse mode, HS09 assumed two modes…"

Further, the deposition scheme has been updated between model versions GEOS-Chem v7.04.13 in HS09 vs. v11-01 used in this work. As discussed in Sect. 3.3 this may have led (in combination with different meteorology) to the different global numbers (about 10%) for the HS09 scheme, compared to the original paper.

3. Page 12, line 8: What molecular weight is used in the model for the fungal spores?

A molecular weight of 31.0 g/mol is applied in the conversion of fungal spore mass from g to gC in the HS09 scheme. We will add a sentence on this.

4. Page 12, lines 14-16: Considering that for the HS09 model, the fungal spores are present mostly in the coarse mode, sedimentation should be a significant process for their atmospheric lifetime calculation. For this, the respective annual budgets (ideally for all model simulations of this paper) should be presented in Table 3 and discussed in more detail in the manuscript. Besides, an additional Table with all simulations performed for this study would be also very useful.

Separating out the dry and wet deposition budgets is easy (readily available from model output). However, sedimentation and dry deposition are not separately available. As in response to comment 2, we will add dry and wet deposition to table 3. This shows that dry deposition is important for the removal of the HS09 coarse mode spores, with

a lifetime against dry deposition of 21 days, while it is twice as long for the fine mode and for the other schemes. Still, wet deposition dominates over dry deposition for the coarse mode spores. Hence the overall lifetime of 1.1 days.

A table with all GEOS-Chem simulations will be added.

5. Page 13, lines 25-26: Do the authors refer here to a new simulation (i.e., with fully emitted land-cover)? If yes, what is the impact on the calculated fungal spores' emissions and burdens? How much would that differ compared to the standard simulation? Is this correction applied only to specific boxes (i.e., those include the coordinates of the observation sites used in this work for model evaluation)?

Yes, we do refer to a new simulation here. However, we don't think it is useful to fully discuss how it impacts emissions and burdens: the only difference between this simulation and the control simulation is that in the simulation with fully emitting land cover, there are spore emissions from water surfaces too. We have done this to make sure that model-measurement comparisons which are situated in a grid box which partly contains water surface do not have an unjustified low bias compared to sites which are in grid boxes with fully emitting (i.e. non-water) land cover. Since the simulation is only included to assure a fair model-measurement comparison (and is not meant to be realistic at a global scale), we see no point in discussing in detail beyond this comparison.
The correction is applied to all boxes, but is only relevant for those boxes that contain a measurement site.

6. Page 17, lines 4-5 & Page 18, lines 1-3: The authors discuss the impact of fungal spores' solubility assumptions (i.e., insoluble vs. fully soluble) on the long-range transport and global burden. Considering, however, a mean lifetime for all simulation of up to ~2 days in the model, the conversion from insoluble to soluble via atmospheric processes (e.g., assuming 1.2-day e-folding conversion from hydrophobic to hydrophilic) may be potentially significant. A short discussion on fungal spores' aging would be here useful.

Atmospheric ageing of fungal spores, and its effect on their solubility, is a process that is subject to large uncertainties. Therefore, we explore the full range of solubilities (from 0 to 1) in Section 4.

We will add the following discussion p17, l. 31:
"Moreover, knowledge on ageing of fungal spores, and the consequences for their behavior in the atmosphere, is limited. Exposure to high relative humidity for several hours may lead to the rupturing of spores, and the formation of cloud-active sub-spore particles (China et al., 2016; Lawler et al., 2020). Further, photo-oxidants, UV-radiation and temperature changes may also induce physical and chemical transformations in bioaerosols (Fröhlich-Nowoisky et al., 2016), potentially altering their solubility."

7. Page 17, line 19: "This suggests that fungal spores contribute less to the organic aerosol budget. . ." Is this statement valid only for PM2.5 spores, or also for spores up to PM10, when the respective emission schemes are applied?

Both, since 1) the new emission schemes are based on measurements which do not have a cut-off size which would exclude spores in the PM10 mode and 2) the size distribution in Fig. 5, which reflects our current best knowledge on spore size distributions in the atmosphere (i.e. with $D_p$=2.5 μm and σ=1.5). The latter does not suggest a bimodal size distribution for fungal spores, while it would still contain significant mass in the coarse mode (with 2.5 <$D_p$< 10 μm).

Technical corrections
i. Page 3, lines 23-28: A more detailed outlook paragraph at the end of Sect. 1 would be useful for the reader.

We will include section numbers here:

'In this study, we develop two new schemes for the emission of fungal spores on seasonal time scales (Section 2), using a previously unexplored source of observed fungal spore concentrations over the United States, and building on available knowledge about the drivers of their emissions. Subsequently in Section 3, we implement these new emission schemes in the GEOS-Chem chemical transport model (Section 3.1) to calculate the global emissions and burden of fungal spores (Section 3.3). Finally, we evaluate the ability of both emission schemes to simulate spatial and seasonal variations in observed fungal spore concentrations and compare results from the new schemes to those from the previously developed Heald and Spracklen (2009) scheme (Section 3.4).'

ii. Page 33: Figure 7 fits better in the supplement.

Since referee 2 asked for further clarification of this figure, we have decided that it would be worth to keep Fig. 7 in the main paper. We will add figures of wet deposition in the supplement to support the interpretation of Fig. 7.

iii. Page 34: Please explain better in the caption the "simulated" vs. "calculated" emission fluxes.

We will update the caption to better reflect that:
Simulated emission fluxes are from GEOS-Chem simulation
Calculated emission fluxes are derived from observations

iv. Page 35: Please explain better how the "normalized" vertical profiles are calculated.

Normalized profiles are obtained by applying min-max normalization, which scales all values to a range between 0 and 1. We will add in the caption.

**Referee #2**
Review of acp-2020-569
Drivers of the fungal spore bioaerosol budget: observational analysis and global modelling

The present study uses a multi-year spore count dataset in order to derive spore number emissions. Two different emission models (a statistical and a population model) are fitted to these derived emissions. Finally, the two emission models are included into the GEOS-Chem aerosol transport model. The resulting spore masses are compared to the spore emission scheme that was already implemented in GEOS-Chem. Large difference were found between the old and the two schemes developed in this study. The three spore emission schemes were evaluated against the dataset that was used to derive the emissions and against ground-level FBAP observations and vertical FBAP profiles. In general, the method to derive spore emissions is scientifically sound and promising for to give a more complete picture if more datasets become available to constrain the emission schemes. Furthermore, I really appreciate the study design involving a variety of different observations for model evaluation. However, in particular the sections discussing the GEOS-Chem results and evaluation (3.3. and 3.4) have some considerable weaknesses when it comes to give reasons and explanations for the findings and differences between the schemes and to give necessary information. For example it is not clear if the time period of the different observations was actually simulated or if the observations were compared to simulations of the year 2016 (and perhaps 2015), which is indicated by section 3.1. At least, I would need more information in order to evaluate the findings of this study. My concerns and questions are addressed in the comments below.

Thank you for the thorough review of our manuscript. We address your comments below.

Major comments
p.4, l.8: How does the sampling rate of 3 days a week relate to the filtering by rain? How long is the collection time in these cases? From the description, I assume 2-3 days. Was the datapoint excluded if rain was found on any of the days of the collection?

We only removed the days on which a fungal spore count and rainfall was reported. This means that if the spore counts at a site was reported at –say- the third day of a 3 day sampling period and rain occurred on the 1st and/or second day of this period, but not on the 3rd day, it was not discarded. Strictly speaking, it should have been. However, this would require a complete reanalysis of the observation data and a repetition of all model runs. Instead, we would like to note here that any rainfall dataset is likely to contain large uncertainties, which may lead to days excluded from the data that should have been left in and vice versa.

p.5, l.6-7: Does this also involve rainfall in the surrounding grid cells?

No, it does not. Each NARR grid box represents an area of 32x32 km$^2$, which we think is fairly representative of the observations made at point locations. Ideally, we would have liked to use rainfall observations at the location of each single AAAAI station, but in absence of that kind of data, we think the NARR rainfall data is fit for purpose.

p.6, l.13-16: I think, the method applied in this work is correct, however I don't understand the reasons given in these sentences. The description of the method is difficult to follow and could be extended by providing more information on certain assumptions. I can't follow the explanation here. Can the authors please explain why short atmospheric lifetimes necessarily lead to smaller concentration differences in the horizontal compared to the vertical? I would have argued that long lifetimes lead to small concentration differences due to longer time for mixing. Since horizontal wind speed is usually orders of magnitude stronger than vertical wind speed, already concentration differences smaller than the assumed here for the vertical (factor of 0.1) may result in the same order of magnitude horizontal advection. Having a read in Bakwin et al., 2004, one can find that they assume horizontal advection to be neglected due to the averaging period of 1month, which perhaps might be the case in regions with no prevailing winds and horizontal relatively homogeneously distributed sources. It's more the long averaging time that smoothes out horizontal differences. By neglecting horizontal advection, the authors implicitly assume long-term horizontal homogeneity, which is also the reason for why only subsidence velocity is applied. I think that this needs to be better described in the text.

This is indeed a confusing sentence, thank you for pointing this out.

'Can the authors please explain why short atmospheric lifetimes necessarily lead to smaller concentration differences in the horizontal compared to the vertical?'
Compared to CO2 one would indeed expect that due to the shorter life time of spores, the horizontal heterogeneity would be stronger, and that consequently the effects of horizontal advection on their concentrations would be stronger too.

We will replace this sentence by:

'For fungal spore concentrations, the horizontal heterogeneity is likely stronger than for a long-lived tracer like $CO_2$, due to the short atmospheric lifetime of these coarse particles and the heterogeneity of their sources. Therefore, horizontal advection possibly has a large influence on spore concentrations. By applying running averages over a period of 20 days, we aim to average out some of this horizontal variability, while acknowledging that this implicitly assumes long-term horizontal homogeneity, which may not be realistic for every AAAAI station.'

p.12, l.32: "...are mainly caused by the occurrence of wet deposition". It sounds reasonable. However, can the authors please show the modelled wet deposition rates similar to Figure 7 or at least modelled rainfall in their answer to this comment? According to p.10, l.17-18, the population model has a "delayed response to temperature and LAI, due to the growth and mortality of the fungi". Can that also play a role here? How long is this delay?

We will include the following figures of wet deposition fluxes in the supplement. GEOS-Chem gives output of wet deposition due to large scale and convective rainfall separately, but here we show large scale wet deposition only, since it is an order of magnitude larger over all continents. They show that the wet deposition over N-America peaks in August.

[Figure]

The delayed response of the population model to temperature and LAI does not play a role here, since it is already accounted for in the timing of the calculated emission fluxes.

p.13, l.5-7: I don't see the strong difference that is described here. Actually, it seems the July and August concentration of statistical and population model are quite similar (0.04-0.05 µg m-3).

Strong is indeed not justified here, so we will remove it. Still, the statistical model emissions show a higher peak than the population model emissions. The concentrations resulting from both models are indeed similar, which is caused by the stronger wet deposition flux for the statistical model spores. We will adapt the text to reflect this.

p.13, l.14: Since wet and dry deposition are the only described loss processes (if I understand correctly), and both loss processes scale directly with the available concentration (i.e. the relative loss is independent of the actual concentration), this statement ("Due to the strong emissions of the HS09 model, emissions and concentrations have similar cycles.") seems wrong. Can the authors clarify what they meant to express?

This is indeed an incorrect statement. Wet deposition is the main removal mechanism, and for HS09 it is assumed that the fine mode spores are in the fine mode (<1 µm) in the wet deposition calculations. This means that they are less efficiently removed from the atmosphere than the spores from the new schemes, which are in the coarse mode in the wet deposition calculations (see figures of modeled wet deposition fluxes). This is the main explanation for the different seasonal cycle in HS09 spore concentrations.
The statement 'Due to the strong emissions of the HS09 model, emissions and concentrations have similar cycles' will be removed from the text.

p.13, l.16-17: Isn't that what the sentence before is saying for the HS09 model? So, I don't understand what different interplay between wet deposition and HS09 and wet deposition and the other two models the authors are targeting at here. For me, the different size assumptions, as explained in the sentence before, seem to be more the cause for different response to the same modelled rainfall rate. However, plotting modelled wet deposition rates would give proof to this speculation.

That is correct. Please see the wet deposition plots above.

p.13, l.25-26: There are a number of stations close to the coast. For me, not necessarily, filling no-emitting water surface with emissions leads to a better comparison between model and observations, since also in the observations in terms of spores presumably clean marine air masses might lead to low concentrations. Furthermore, also the observational sites experience spores from probably a variety of land covers in their surrounding. Can the spore counts really be estimated to be dominated by local sources without a strong contribution of long-range transport (at least from around 200km)? Based on the lifetime shown in Table 3, long-range transport over this distance is occurring frequently. Can the authors please show a comparison of their "fully emitting land cover" simulation and the simulation using the original land cover at least for the stations whose land cover has been changed by this assumption? If I understood correctly, the original land cover was used for the results shown in e.g., Fig. 6.

The reviewer raises a good point, and as discussed above, the role of horizontal advection is challenging to ascertain for these sites. So there is no way to guarantee that the spore counts are dominated by local sources. In the supplement, we will add the figures for the control simulation, which will show that the model-measurement agreement is worse than for the fully-emitting land use simulation

p.13, l.26: Does "fully emitting land cover" also involve grid cells containing nothing but ocean? From the text, this is not clear.

It does not, so we will clarify this.

p.13., l.27-29: I would have guessed that in particular the variability of meteorology in different years is one of the main causes of differences. Since different years were observed and simulated, how is the correlation coefficient calculated, i.e. which timeseries is/are or population mean are compared against each other? On p. 14, l.5, low "skill in reproducing seasonal variations at the AAAAI stations" is determined by small correlation coefficients. Even for long-term averages such as seasons, the meteorological driving variables might be different in different years. Can the authors please show a comparison of the 2016 meteorological conditions and the observational years?

'Since different years were observed and simulated, how is the correlation coefficient calculated, i.e. which timeseries is/are or population mean are compared against each other?'

Simulated emissions from 2016 simulation are compared to averaged observations over the years 2003-2008. We compare monthly mean values for each station. The figures below show the seasonal cycle of the meteorological variables that drive the modeled spore emissions, i.e. 2m-temperature, 2m-specific humidity and friction velocity. They show that there is good agreement between 2016 meteorology and the mean meteorology over 2003-2008, with low bias and correlation coefficients above 0.9 for temperature and humidity. Correlation coefficient is lower for friction velocity, but this is the least important variable in the statistical model.

[Figure]

*Figure 1: comparison of MERRA2 meteorology used to drive the GEOS-Chem simulations and NARR meteorology at each AAAAI station. MERRA2 seasonal cycle for 2016 is compared to mean NARR data over 2003-2008. Shaded areas show standard deviations. From top to bottom: 2 m temperature, 2 m specific humidity and friction velocity.*

p.13, l.34: The 25% / 75% share between fine and coarse mode is reported for the mass in the abstract of Heald and Spracklen (2009). So, for spore mass, this statement absolutely makes sense. However, in Fig. 8, I think the number emission flux is shown, right? Does the HS09 scheme really show a larger number of coarse mode spores than number of fine mode spores? Heald and Spracklen (2009) actually state "...we obtain number concentrations of 10ˆ5 m-3 (fine) and 8 x 10ˆ3 m-3 (coarse)." (p. 4, paragraph [9]). This does not seem to be inconsistent with the observed size distribution, which the authors refer to in the next sentence. Furthermore, it suggests that the overestimation is either general (i.e.,

both in fine and coarse mode) due to the applied method by Heald and Spracklen (2009) or at least in the fine mode. Can the authors please clarify what they meant to express here?

Fig. 8 indeed shows the number emission flux, and for the HS09 model, only the fine mode numbers are shown. We will add this information in the caption. Adding the HS09 coarse mode number emission on top of that would lead to an even larger overestimation.

To distinguish better between the emitted number and mass and the contribution of the fine and coarse modes, respectively, we will replace:
'The HS09 scheme also reproduces this pattern, but with a strong overestimation of emissions over the whole US (NMB=10.1), as expected given the order of magnitude difference in emissions. We note that this overestimate is largely driven by the inclusion of the coarse mode emissions in HS09, which make up 75% of the emissions based on the mannitol observations used to constrain that model.'

By:
'The HS09 scheme also reproduces this pattern, but with a strong overestimation of number emissions over the whole US (NMB=10.1), even when looking at fine mode spores only. This overestimate of number emissions is expected given the order of magnitude difference in emitted mass. We note that while the overestimate of emitted mass is largely driven by the inclusion of the coarse mode emissions in HS09, which make up 75% of the emissions based on the mannitol observations used to constrain that model, the overestimate in emitted spore numbers is mainly due to emissions in the fine mode.'

Section 3.4: I assume that the observed FBAP concentrations are from other years than 2016. Are the same years simulated or is the 2016 simulation used instead? This is not stated in the text and should be mentioned. In section 3.1 it is written that only the years 2015 and 2016 were simulated. If other simulated time periods than the observed ones were used, the knowledge gain from the results presented in this section is limited. I would not necessarily expect good comparison for single peaks in the seasonal cycle and timeseries statistics as different meteorological conditions and timing can vary greatly between different years. The section should include clear statements on these limitations if other years were simulated than observed. In particular it should then be checked how the meteorological variables that drive the emissions compare at or close to the sites between the different years.

For all comparisons, we used the 2016 simulation. FBAP datasets are from different years indeed, and we will add more information on the dates of the campaigns that in the text. For the NAAMES 2016 period, year of measurement and year of simulation agree.

p.15, l.15-16: Can the authors clarify what is meant here? For me, it looks like that at Karlsuhe (which I believe the statement is referring to) HS09 and observations peak right at the same time in June, however HS09 is just overestimating. In case you mean that the maximum concentration in HS09 is in June whereas it is Aug-Oct in the observation, I recommend to revise this sentence.

We will revise the sentence

p.16, Section on the comparison to vertical profiles of FBAP: Were the same time periods simulated in which the obersvations took place?

Only for the NAAMES 2016 campaign, which happened to be in the simulated year. We will state this in the text

p.16, l.15-16: Which modelled time period was compared to these observations?

Same period in year, but for the year 2016

p.17, l.2-3: How does this compare for the other years with observations (2015 and 2017)? At least, 2015 was simulated.

2015 was only used for spinup, so we cannot use this output for model-measurement comparisons.

p.17, l.17-18: "These differences are largely the result of different assumptions about size...". If I understood correctly, the HS09 is emitting the mass. So size does not play a dominant role for the mass concentration. Is the HS09 emission in GEOS-Chem implemented as mass- or number-based emission scheme?

Size matters in HS09, since there are 2 modes. In the fine mode, the number emission is higher than in the coarse mode, but in the fine mode the emitted mass is smaller than in the coarse mode.
In GEOS-Chem it is implemented as mass-based emission scheme. Mass concentrations for the fine and coarse modes are converted to number concentrations by applying a spore mass of $1.0 \times 10^{-12}$ and $1.3 \times 10^{-10}$ g, respectively. These numbers are based on monodisperse distributions with geometric diameters in the middle of the size distribution for the fine and coarse modes and a density of $1 \times 10^6$ g/m$^3$.

Other comments

p.4, l.10: What does "no local spore or pollen sources" mean? How close is local in this context?
This comes from the description at the AAAAI website. No specifics are given there on how close local means. Since it is not clear what it exactly means, we will remove this part of the sentence.

p.5, l.17: Do you mean subsidence of cold air? If not, then please further explain what is meant here.
'Free-tropospheric' air reflects better what we mean here than warm air, so we will change it

p.6, l.4: The abbreviations BL and FT have not been introduced, yet.
Will introduce them here

p.6, l.7: I'm not sure if the abbreviation for local time is well known.
Will write out 'local time' here, since it is not used elsewhere in the text

p.6, l.9-10: I assume that the boundary layer height is also taken from NARR data? Further, the authors write "mean height of the afternoon boundary layer", however, in the Figure caption of Fig. 2 they write "c) maximum daily boundary layer height". Can the authors please clarify, which was used?
BL height is taken from NARR indeed; will clarify in text.
The mean height of the afternoon boundary layer is used. Will clarify this in the caption of Figure 2

p.6, l.10: Double "daytime" in "the daytime mixed-layer during daytime".
Removed first 'daytime'

p.6, l.33-34: How was this calculated?
By applying the offline version of the dry deposition scheme as mentioned in l. 29-30

p.8, l.23-25: This sentence is rather complicated. It could perhaps be split into two sentences for easier reading?
Will split sentence into:
'Rather, they are variables which show a similar seasonal cycle as, and therefore a statistical relationship with, the emissions over all stations and years. Therefore, they can be tentatively associated with the growth of fungi and the emission of spores.'

p.8, l.31: I think, referring to section 2.5 where the fitted parameters are presented or even to Table 2 already in the beginning of section 2.4 is beneficial for the reader here.
Will add reference to section 2.5

p.9, l.22: Perhaps nothing to bother before type setting, but the font seems to have changed.
Seems a different font indeed, but will leave it for the type setting

p.10., l.17-18: Double "in the population model".
Will remove the second 'in the population model'

p.10., l.25: Perhaps name it "derived spore emissions" to better make clear that the observed emissions are meant.
Will do

p.11, l.5: Can the authors please show the comparison of LAI for 2008 and 2016 in an answer to this comment?

The figure below shows the comparison between the LAI for 2008 and 2016. 2008 LAI as applied in GEOS-Chem is shown on the x-axis and the 2016 LAI on the y-axis.

[Figure]

p.12, l.10: For me, burden is usually a column quantity or mass / number in the whole atmosphere. However, shown in Figure 7 is a concentration [µg m-3] according to the axis label. I therefore recommend to use "concentrations" instead of "burden", as it is done later in this section.
We will use 'burden' when we refer to total mass in the atmosphere, and 'concentration' when we refer to more local numbers. Will check the MS and apply consistently.

p.13, l.18-19: This sentence reads a bit difficult. First, at this point, it is not clear (since it is the beginning of a new paragraph) that with "both schemes" the authors mean statistical and population model. Second, the fragment "as well as the HS09 scheme" is badly placed / written for easy understanding.
Will change sentence into:
'As a verification of our implementation of the statistical and population emission schemes, we compare the results of both schemes within the GEOS-Chem simulation to the AAAAI data from which they were developed. In addition, we also compare the results from the HS09 scheme as implemented in GEOS-Chem to the AAAAI data.'

p.13, l.19-20: I get your point, but written this way, it seems like a rather strong statement. I would have not assumed it, especially since a different time period is simulated (2016) than observed (2003-2008).

We indeed mention in l. 28-29 that the different time period contributes to the difference between model and observations.

p.13, l.30: The reference to Figure 8 should already be given somewhere in the beginning of this paragraph.
Will add figure reference in l. 30

p.14, l.6: Just for my understanding. The comparison and correlation coefficients in Fig. 4 are for the timeseries of the emission mean of all stations? If so, I suggest to write this clearer in the respective section.
The comparisons are indeed for the mean of the emissions over all stations. On p10, l10, we will change 'the calculated emission time series for each individual station' to 'the mean calculated emission time series over all stations'.

p.15, l.24: Actually, the population model stays rather low. I would not call the increase "sharp" for this particular model.
Will change 'a sharp increase' into 'an increase'.

p.15, l.29: Why was the temperature threshold not introduced for the HS09 model?
Because we decided to stick to the original HS09 model as closely as possible.

p.18, l.8: Please consider revising this statement ("continental outflow of bioaerosols"). Only spores were simulated.
Since the observations that are dealt with here are of FBAP, we will change 'bioaerosols' into 'FBAP'.

Table 1: Assuming [q] = g/g (or similar) and [LAI] = m2m-2 the unit of b1 and b2 should be m2s-1?
Thanks for catching this. Will adopt the appropriate units here.

Table 3: Typo: "Table. 3" -> "Table 3".
Will change

Figure and Table captions: Probably not something to bother at this stage: Sometimes dots are missing at the end and non-capital letters in the beginning of Figure and Table captions, in both the manuscript and the supplement.
Will catch this in type setting

Figure 2: a-e missing in the Figure itself, but it is referred to in the text and the figure caption.
Will add

Figure 3: In the Figure caption, "2" is not in superscript in rˆ2.
Will change

Figure 4: Perhaps typo in the figure caption (I'm not a native speaker)? "20-day running derived-emission flux" -> "20-day running mean derived emission flux".
Will change

Figure 4: "fit parameters shown inset". This is not the case. However, it is not needed since fit parameters are presented in Table 1 and 2 very nicely.
Indeed it's not the fit parameters that are show inset, but rather the statistics describing the comparisons. Will change this.

**Referee #3**

Overall comments:
This manuscript presents the development of a new parameterization, suitable for use in regional and global atmospheric models, of the emissions of fungal spores to the atmosphere, as a function of meteorological and land surface parameters. The new parameterization is derived based on a large dataset of fungal spore counts from the American Academy of Allergy, Asthma and Immunology (AAAAI), which has previously not been exploited for this purpose. Since visual counts of fungal spores are widely understood to be the most reliable measurement of atmospheric fungal spore concentrations that is typically available (despite potential limitations), a parameterization based on this new data source can be expected to have greater reliability than previous parameterizations based on other proxy measurements (e.g., mannitol concentrations, and concentrations of fluorescent biological aerosol particles, FBAP). The new parameterization should especially be relevant within the region from which the observational data were obtained (North America), but has been developed on the basis of variables that are globally available from observational datasets and/or within atmospheric models.
In addition, the new parameterization uses an approach to estimating the relationship between fluxes and near-surface concentrations based in a simplified approach to modelling the convective boundary layer that involves some limitations and assumptions, but which is more sophisticated than (and likely an improvement upon) the approaches taken in the development of some earlier parameterizations for fungal spore emissions. The parameterization is selected via a regression model, which is similar to the approach taken in Heald and Spracklen (2009), but which considers more variables and uses an improved statistical approach for model selection (i.e. multiple linear regression with model

selection via the Bayesian information criterion to select the best model while avoiding over-fitting). Also, a biological-growth-based model is proposed in addition to the statistical regression model. Finally, the new parameterization is evaluated by comparison with normalized FBAP measurements (seasonal cycles and vertical profiles), and several sensitivities of the model are discussed.

In summary, this paper represents a significant advance in emissions modelling of fungal spores, and is within the scope of Atmospheric Chemistry and Physics. Most of the questions I had are already addressed by the authors with appropriate caveats in the manuscript in its current form. The neglect of horizontal advection in the inference of emission fluxes is likely a meaningful limitation, but one that is not possible to address with the approach/framework used here. Diurnal cycles of emissions (and their interaction with the diurnal cycle of the convective boundary layer) are also not addressed, but it appear that the existing data do not have sufficient time resolution to allow investigation of these cycles. Based on my evaluation, I recommend that it be published after the following questions and comments are addressed.

We would like to thank Referee #3 for the constructive remarks on our manuscript. We will revise the MS according to these comments as described below:

General questions and comments:

1. The main question I had about this paper is regarding the equilibrium boundary layer approach used to derive the flux estimates. I was not entirely convinced that the prior use of this method for inferring $CO_2$ fluxes is adequate justification for its use in inferring aerosol fluxes, since $CO_2$ is considerably more well-mixed in the atmosphere and has fewer complicating removal processes (especially wet removal). The study by Perring et al. (2015) is cited as showing that FBAP concentrations decline with altitude within the PBL, which seems to contradict the reliance on the assumption of well-mixedness. The approach relies on the assumption that convection maintains a well-mixed boundary layer; this assumption will not always be met, and there are likely systematic relationships between the times when the assumption is violated and some of the model's predictor variables (e.g., near-surface temperature). Diurnal cycles in emissions could also complicate the validity of the approach.

I think some discussion/analysis of how frequently the underlying assumptions of this approach are likely to hold would be warranted – especially the assumption of a boundary layer that is well-mixed with respect to both scalars and aerosols.

This is a valid comment, but also one that is hard to address. Few vertical profiles are available to test assumption of well-mixed profile for aerosols in the PBL, and these are mostly limited to flight campaigns which are limited in their spatial and temporal scope (Figure 7 in Twohy et al. (2016) shows at least 1 day with a well-mixed FBAP-profile). Vertical profiles of potential temperature (available from reanalysis data) could give an indication whether the assumption of a well-mixed PBL is valid, at least for scalars, over larger areas and longer time periods. (However, the reliability of profiles from reanalysis data ultimately comes down to the boundary layer scheme used in the model used for the reanalysis.)

However, we needed to select a method to perform the inversion from concentration to emission. Several options are available, each one with its own limitations. An inversion based on GEOS-Chem simulations would, for instance, have suffered from similar limitations, since its standard PBL scheme assumes a well-mixed boundary layer under any condition. Using the mixed-layer assumption allowed us to run some sensitivity tests to evaluate what the effects are of the choice of parameters for vertical mixing, which we explore in Section 4. We used 20-day running means to average out effects of diurnal variations in boundary layer dynamics to some extent.

Finally, we can look at the potential effects on our conclusions. It seems likely that concentrations of spores are highest close to sources, i.e. near the surface (as shown by Perring et al., 2015). So the effect of taking these concentrations as representative of boundary layer values, is that we overestimate the emission fluxes. This would mean that calculated emissions at the global scale should be regarded as upper limit values.

We will add (p. 5, l. 21):
"We have to note here that it is hard to assess the validity of the assumption of well-mixed profiles of fungal spores in the boundary layer, since only limited observations of vertical profiles throughout the boundary layer are available. Observations show that concentrations of spores are actually highest in the surface layer (Perring et al., 2015) where the

AAAAI measurements are taken. Taking these concentrations as representative of boundary layer values means that we overestimate their emission fluxes. Calculated emissions in this work should therefore be regarded as upper limit values. We explore the sensitivity of these emissions to assumptions on vertical mixing parameters in Section 4."

2. It strikes me as almost slightly contradictory that the temperature plays such a small role in the statistical model obtained via linear regression (Figure 3), yet the threshold value in temperature is shown to have a large impact on simulated emissions, and temperature also is a key variable in the population model. A priori, I would expect that fungal spore growth has an important, but non-linear, dependence on temperature, where growth would be inhibited at colder temperatures that are sub-optimal for fungal spore growth (as is also embodied in the population growth model). I wonder if the model would show a dependency on T if the analysis were repeated with a different statistical (or machine learning) method that allows for potential nonlinear dependencies.

I recognize that would entail a significant amount of work (essentially repeating the entire study), which is not necessary (and might not lead to improvement!). But here I think it would be helpful if the authors could comment on (1) whether such approaches were tried and discarded for some reason, and (2) whether there is any notable relationship between the model-data mismatches (in modelled versus derived emissions and likely predictor variables including T at 2m and 10m (as might be revealed by a scatterplot).

1. Such approaches were not tried. We think that the fact that we developed two emission models already acknowledges to some extent the limitations of the statistical model in terms of accounting for non-linear relations. Still, we see that both models perform similarly a) in comparison with the AAAAI emissions, b) on the global numbers and c) in comparison with the FBAP data. A third approach in which a non-linear statistical method would have been applied may have led to somewhat different results, but it seems unlikely that it would have affected the main conclusions of this work.
2. We made a few scatterplots, but found no notable relationships between model-measurement differences and predictor variables. Below are shown plots for statistical model emissions and T2m and T10m, respectively. Correlation coefficients are very small (~0.004). Each data point represents a monthly average value for a single AAAAI station.

[Figure]

[Figure]

3. The normalization of FBAP to compare with the spore data is appropriate considering the limitations of both types of observations. However, I think the normalization factors should be reported, as it would be informative for readers to know how much scaling had to be applied and how consistent or different this was between the datasets. Additionally, for the normalized vertical profiles in Figure 10, I was unable to find an explanation in the text of how the normalization factor was determined (I think so that the largest value in each vertical profile is 1?).

The normalization factors are as follows:

| Campaign | Normalization factor |
|---|---|
| Germany | 9.6E-3 |
| Finland | 0.014 |
| Colorado | 0.019 |
| Ideas | 0.095 |
| SEAC4RS | 0.0036 |

| NAAMES 2015 | 0.15 |
|---|---|
| NAAMES 2016 | 0.048 |
| NAAMES 2017 | 0.12 |

The table shows that for the ground-based observations, the normalization factors are within a narrow range. For the flight campaigns, all normalization factors are within a factor ~3 from each other, with the exception of SEAC4RS, for which the factor is an order of magnitude lower. We will add this table in the supplement.
In the time series (Fig 9) and vertical profiles (Fig 10) we applied min-max normalization, which scales all values to a range between 0 and 1 (see https://en.wikipedia.org/wiki/Feature_scaling). We will clarify this in the text.

4. A key difference between the new proposed scheme and the HS09 scheme, which I think is not discussed, is the geographic representativeness. The mannitol data used in the HS09 scheme (Elbert et al., 2007; Table A3) includes a large number of data points from tropical rainforests of Brazil, which are not represented in the AAAAI dataset, as well as some extratropical data, which are mostly from Europe. It should be pointed out explicitly to readers that the geographic sampling is quite different from the data used for the previous parameterization (in addition to the differences in the measurement type and assumed size distribution, which are already noted).

Will add some discussion in Section 4.

p.17, l.19
'Additionally, the data on which the HS09 scheme was developed contained a large number of data points from tropical forest, which are absent in the AAAAI dataset. This means that the simulated emissions over tropical forests from the statistical and population model are in fact extrapolations based on data from temperate ecosystems.'

p. 17, l. 19-20
Moved up to l. 16 and slightly modified the line 'This suggests that fungal spores contribute less to the organic aerosol budget of the atmosphere and are likely less important for cloud and precipitation formation than previously estimated in models.' to keep the flow of the text.

Minor and typographical comments:

P 6, l. 18-19 and l. 23-24 are partially redundant.
Not sure which parts are meant here.

p. 7, l. 24: some commas missing here inside the parentheses
Will add some more parentheses, rather than commas, for clarification: '(temperature at 2 m ($T_{2m}$) and specific moisture at 2 m ($q_{2m}$))'

References

China, S., Wang, B., Weis, J., Rizzo, L., Brito, J., Cirino, G. G., Kovarik, L., Artaxo, P., Gilles, M. K. and Laskin, A.: Rupturing of biological spores as a source of secondary particles in Amazonia, Environ. Sci. Technol., 50(22), 12179–12186, https://doi.org/10.1021/acs.est.6b02896, 2016.

Elbert, W., Taylor, P. E., Andreae, M. O. and Pöschl, U.: Contribution of fungi to primary biogenic aerosols in the atmosphere: wet and dry discharged spores, carbohydrates, and inorganic ions, Atmospheric Chem. Phys., 7(17), 4569–4588, https://doi.org/10.5194/acp-7-4569-2007, 2007.

Fröhlich-Nowoisky, J., Kampf, C. J., Weber, B., Huffman, J. A., Pöhlker, C., Andreae, M. O., Lang-Yona, N., Burrows, S. M., Gunthe, S. S., Elbert, W., Su, H., Hoor, P., Thines, E., Hoffmann, T., Desprès, V. R. and

Pöschl, U.: Bioaerosols in the Earth system: Climate, health, and ecosystem interactions, Atmospheric Res., 182, 346–376, https://doi.org/10.1016/j.atmosres.2016.07.018, 2016.

Heald, C. L. and Spracklen, D. V.: Atmospheric budget of primary biological aerosol particles from fungal spores, Geophys Res Lett, 36, 2009.

Lawler, M. J., Draper, D. C. and Smith, J. N.: Atmospheric fungal nanoparticle bursts, Sci. Adv., 6(3), eaax9051, https://doi.org/10.1126/sciadv.aax9051, 2020.

Perring, A. E., Schwarz, J. P., Baumgardner, D., Hernandez, M. T., Spracklen, D. V., Heald, C. L., Gao, R. S., Kok, G., McMeeking, G. R., McQuaid, J. B. and Fahey, D. W.: Airborne observations of regional variation in fluorescent aerosol across the United States, J. Geophys. Res. Atmospheres, 120(3), 1153–1170, https://doi.org/10.1002/2014JD022495, 2015.

Twohy, C. H., McMeeking, G. R., DeMott, P. J., McCluskey, C. S., Hill, T. C. J., Burrows, S. M., Kulkarni, G. R., Tanarhte, M., Kafle, D. N. and Toohey, D. W.: Abundance of fluorescent biological aerosol particles at temperatures conducive to the formation of mixed-phase and cirrus clouds, Atmospheric Chem. Phys., 16(13), 8205–8225, https://doi.org/10.5194/acp-16-8205-2016, 2016.